# Distinct dissociation rates of murine and human norovirus P-domain dimers suggest a role of dimer stability in virus-host interactions

Robert Creutznacher[1,8], Thorben Maass[1], Jasmin Dülfer [2], Clara Feldmann[1], Veronika Hartmann [3], Miranda Sophie Lane[3], Jan Knickmann [3], Leon Torben Westermann[1], Lars Thiede[2,4], Thomas J. Smith[5], Charlotte Uetrecht[2,4,6], Alvaro Mallagaray [1], Christopher A. Waudby [7], Stefan Taube [3✉] & Thomas Peters [1✉]

Norovirus capsids are icosahedral particles composed of 90 dimers of the major capsid protein VP1. The C-terminus of the VP1 proteins forms a protruding (P)-domain, mediating receptor attachment, and providing a target for neutralizing antibodies. NMR and native mass spectrometry directly detect P-domain monomers in solution for murine (MNV) but not for human norovirus (HuNoV). We report that the binding of glycochenodeoxycholic acid (GCDCA) stabilizes MNV-1 P-domain dimers (P-dimers) and induces long-range NMR chemical shift perturbations (CSPs) within loops involved in antibody and receptor binding, likely reflecting corresponding conformational changes. Global line shape analysis of monomer and dimer cross-peaks in concentration-dependent methyl TROSY NMR spectra yields a dissociation rate constant $k_{off}$ of about $1 \, s^{-1}$ for MNV-1 P-dimers. For structurally closely related HuNoV GII.4 Saga P-dimers a value of about $10^{-6} \, s^{-1}$ is obtained from ion-exchange chromatography, suggesting essential differences in the role of GCDCA as a cofactor for MNV and HuNoV infection.

[1] University of Lübeck, Center of Structural and Cell Biology in Medicine (CSCM), Institute of Chemistry and Metabolomics, Ratzeburger Allee 160, 23562 Lübeck, Germany. [2] Leibniz Institute of Virology (LIV), Martinistrasse 52, 20251 Hamburg, Germany. [3] University of Lübeck, Center of Structural and Cell Biology in Medicine (CSCM), Institute of Virology and Cell Biology, Ratzeburger Allee 160, 23562 Lübeck, Germany. [4] CSSB Centre for Structural Systems Biology, Deutsches Elektronen-Synchrotron DESY, Notkestraße 85, 22607 Hamburg, Germany. [5] University of Texas Medical Branch at Galveston, Department of Biochemistry and Molecular Biology, 301 University Boulevard, Route 0645, Galveston, TX 77555, USA. [6] School of Life Sciences, University of Siegen, 57076 Siegen, Germany. [7] Institute of Structural and Molecular Biology, University College London, Gower Street, London WC1E 6BT, UK. [8] Present address: Virology Division, Department of Infectious Diseases and Immunology, Faculty of Veterinary Medicine, Utrecht University, Yalelaan 1, 3584 CL Utrecht, The Netherlands. ✉email: stefan.taube@uni-luebeck.de; thomas.peters@uni-luebeck.de

Noroviruses are non-enveloped, positive-strand RNA viruses, belonging to the family of *Caliciviridae*. Human noroviruses (HuNoVs) are responsible for more than 600 million cases of viral gastroenteritis worldwide annually[1–3] and pose a substantial burden to healthcare systems with no licensed vaccines or antiviral therapies currently available. Novel cell culture systems for HuNoV have been a major advancement in the field but they are still somewhat limited in that they are not trivial to implement and depend on infectious viruses obtained from patient samples[4,5]. Murine noroviruses (MNV), on the other hand, share the enteric tropism with their human counterpart and can be easily studied in cell culture and their native small animal host[6,7]. Noroviruses share a common capsid structure containing 180 copies of the major capsid protein VP1 forming $T = 3$ icosahedral particles. Each VP1 is composed of an N-terminal shell (S)-domain and a C-terminal protruding (P)-domain. With the overall capsid structures of HuNoV and MNV being similar one might expect comparable viral strategies to enter host cells. However, important differences exist, particularly during host cell entry[8]. For instance, MNV uses the murine membrane glycoproteins CD300lf or CD300ld as proteinaceous entry receptors[9,10], whereas HuNoV strains instead require histo-blood group antigens (HBGA) for infection. Interestingly, both HuNoV and MNV bind glycochenodeoxycholic acid (GCDCA)[11,12]. Although GCDCA binds to HuNoV and MNV capsids at very different locations and with large differences in binding affinity, GCDCA can promote infection in either species, for HuNoV in a strain-specific manner[5,11–13]. Exciting results from cryo-EM studies showed that norovirus capsids exist in at least two major forms, one with the P-domain hovering 10–15 Å above the shell domain representing an extended conformation, and another one with the P-domain tightly associated with the shell surface representing a contracted conformation[14–19]. For MNV, bile salts were key mediators to drive the capsids into the contracted form, with GCDCA being the most effective[15].

Crystal structures of MNV P-dimers complexed with the receptor CD300lf in the absence (PDB 6C6Q) and presence (PDB 6E47) of GCDCA are available[11]. Alignment of the P-domains of the two crystal structures using the program PyMOL[20] yielded a $C_\alpha$-RMSD of 0.132 Å, showing that the structures are virtually identical. The cryo-EM structure of the MNV virion complexed with GCDCA revealed a rotation of the P-domain relative to the S-domain forming a contracted particle, where the P-dimer has collapsed onto the shell surface compared to the apo structure[15,19]. Interestingly, one crystal structure (PDB 3LQ6[21]) differs from all other available MNV-1 P-dimer crystal structures. Only in this structure of the apo P-dimer, the A'B' loop (aa 299–301) and the E'F' loop (aa 379–388) both are found in two different conformations, an "open" and a "closed" conformation[21] as illustrated in Fig. S1. In the open conformation, the C'D' loop (aa 342–351) is lowered and blocks access to the GCDCA binding site at the lateral dimer interface[11]. The binding of GCDCA locks the complex into the closed conformation by pushing the C'D' loop up against the E'F' loop. Structural data further suggest that neutralizing antibodies specifically require the open conformation, while the closed conformation may be more adaptable for receptor recognition[14,22]. This is supported by functional data, showing that MNV complexed with GCDCA abrogates neutralizing antibody (A6.2, 2D3, and 4F9) binding while moderately improving infection[22,23].

Antibody escape mutations at the tip of the A'B' and E'F' loop have been described for the monoclonal antibodies (mAb A6.2, 2D3, and 4F9). Cryo-EM structures suggested that A6.2 and 2D3 mAbs bind to approximately the same location on the MNV P-domain and contact the E'F' loop. Escape mutations to mAb A6.2 lie on the outermost A'B'/E'F' loops (A382K/R, D385E,

V378F, L386F; Fig. S1) and are immediately adjacent to the bound A6.2 antibody[15,24]. In contrast, the only isolated escape mutations to 2D3 and 4F9 lie buried within the P2 domain dimer on the C'D' loop (D348E and V339I), well away from the bound 2D3 (Fig. S1)[15,24]. Therefore, these latter escape mutations are thought to act in an allosteric-like manner in the viral capsid[24]. This allosteric-like escape mechanism is supported by molecular dynamics assisted flexible fitting simulations that suggested escape mutations such as V339I stabilize the P-domain dimer interface, thereby affecting conformational features of the P-domain and mAb 2D3 and 4F9 binding. Interestingly, none of the natural escape mutations to A6.2 were functional for 2D3 or 4F9 and suggest substantial differences in antibody/virus contacts in spite of having very similar paratope footprints.

The binding of bile acid (GCDCA) is thought to act in a mechanism similar to these allosteric-like escape mutations by shifting the P-domain conformation equilibrium away from the "open" conformation by stabilizing the "closed" state[22]. Furthermore, the binding of GCDCA leads to a different relative orientation of monomeric units within the P-dimer[15], allowing the P-dimer to rotate and contract onto the S-domain without steric clashes[22]. At the same time, the A'B' and E'F' loops reorient such that neutralizing antibodies cannot bind to the contracted form of MNV. From a structural point of view, the different relative orientations of P-domains in the dimer in the absence and presence of GCDCA may explain the transition to the contracted form without steric clashes. However, details of underlying P-domain interactions are largely unknown. Therefore, we have used NMR, native MS, and other biophysical techniques to shine a light on these P-domain interactions and their modulation by GCDCA in the solution state.

## Results

**Synthesis of MNV P-domains and purification.** Most experiments in this study were performed using the P-domain of MNV-1 (CW1)[21]. Additional experiments employed P-domains of MNV strains CR10[11], and MNV07[25], sharing more than 90% sequence identity with CW1 (Fig. S2 and Table S1). Crystal structure data are available for several P-domains including the strains CW1 and CR10[11,21,26], allowing correlation with data from NMR binding experiments with stable isotope-labeled proteins[27–31]. Compared to prior work[32], we used P-domain constructs that were truncated at the C-terminus. Our standard protocol for protein biosynthesis and purification as established for HuNoV P-domains[33,34] led to irreversible unfolding and protein aggregation of MNV P-domains (Fig. S3a). We found that MNV P-domains were labile at neutral pH but purification in acidic conditions prevented aggregation and provided properly folded protein (Fig. S3b). Analysis of the thermostability of P-domains underlines the importance of adjusting the pH since the relative stability quickly decreases with increasing pH values above pH 6 (Fig. S3c). The modified protocol was further optimized for the preparation of [$U$-$^2$H,$^{15}$N] labeled and specifically $^{13}$C-methyl (MIL$^{proS}$V$^{proS}$A) labeled MNV CW1 P-domain samples, allowing $^1$H,$^{15}$N TROSY HSQC, and methyl TROSY-based chemical shift perturbation experiments, respectively, into P-domain dimerization and ligand binding.

**NMR experiments uncover concentration-dependent monomer-dimer interconversion.** Recording a series of $^1$H,$^{15}$N TROSY HSQC spectra of [$U$-$^2$H,$^{15}$N] labeled CW1 P-domain samples for protein concentrations ranging from 25 to 200 µM yields spectra with obvious concentration-dependent changes. Besides the overall increase in backbone H$^N$ cross-peak intensities due to improved signal-to-noise ratios at higher concentrations,

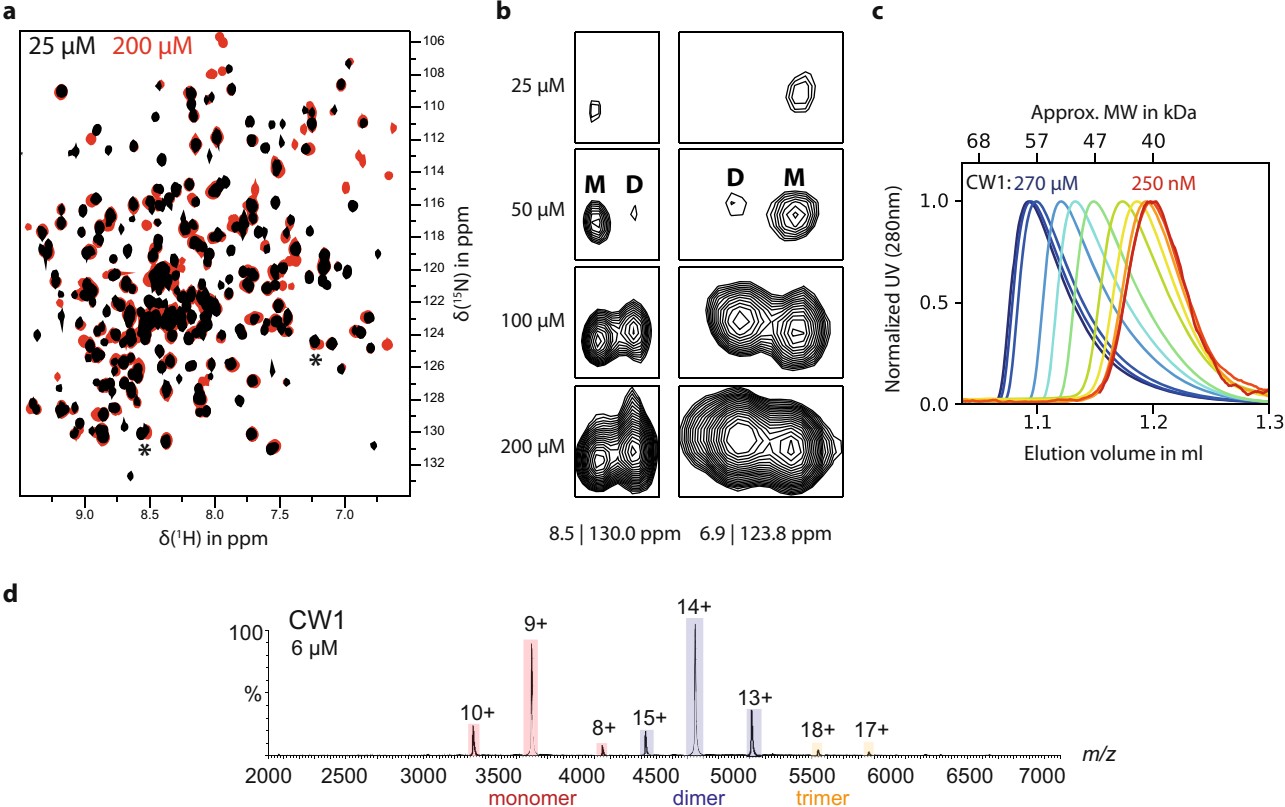

**Fig. 1 Concentration-dependent dimerization of the MNV P-domain in solution. a** $^1$H,$^{15}$N TROSY HSQC spectra of [$U$-$^2$H,$^{15}$N] labeled MNV CW1 P-domains show protein concentration-dependent changes (black: 25 μM, red: 200 μM). Several new backbone NH signals appear with increasing protein concentration while others decrease in relative intensity. **b** These signals can be assigned to originate from the dimer (D) or the monomer (M), respectively. Two selected signal pairs are shown (marked by asterisks in **a**). **c** CW1 P-domains have been subjected to size-exclusion chromatography at increasing concentrations, ranging from 250 nM (red) to 270 μM (blue). The elution volume strongly depends on protein concentration and can be associated with an increase of apparent molecular weight (Fig. S4) from 39.6 kDa (monomer) to 58.4 kDa (dimer). The normalized chromatograms reveal a noticeable peak tailing at higher concentrations, indicating ongoing monomer-dimer exchange on the time scale of the chromatographic separation. **d** Native mass spectra of CW1 P-domains confirming monomer-dimer equilibrium.

there are two sets of cross-peaks with concentration-dependent relative intensities. New cross-peaks emerge with increasing P-domain concentrations. Concomitantly, another set of cross-peaks decreases in relative intensity as is exemplified in Fig. 1a, b. Concentration-dependent behavior is also noticeable in complementary biophysical experiments. Size-exclusion chromatography (SEC) analyses with P-domain concentrations ranging from 250 nM to 270 μM reflect an increasing apparent molecular weight at higher concentrations (Fig. 1c and Fig. S4), and native mass spectrometry (MS) experiments directly show that CW1 P-domains exist as a mixture of monomers and dimers (Fig. 1d).

Normalized SEC curves show increased tailing at higher concentrations indicating monomer-dimer exchange on the time scale of the chromatographic separation[35]. Multi-angle light scattering (SEC-MALS) confirms that monomers dominate the tailing part of the chromatographic peak, whereas the front part consists mostly of dimers (Fig. S5). Therefore, the two sets of cross-peaks in the HSQC spectrum reflect an equilibrium between monomeric and dimeric P-domains. It follows that the peaks increasing in intensity with increasing protein concentration must be assigned to P-domain dimers, and the peaks decreasing in intensity to P-domain monomers.

To determine a dissociation constant $K_{D,Dimer}$ for the P-domain monomer-dimer equilibrium we used NMR, native MS, and SEC. From native mass spectra a $K_{D,Dimer}$ value of 32 μM is estimated (Fig. S6c). Quantitative analysis of SEC profiles is possible but is affected by the inherent dilution step when the protein enters the resin from the sample loop directly influencing this equilibrium. Nevertheless, fitting the law of mass action to SEC data and taking into account the uncertainty of the actual protein concentration yields a range of 4–40 μM for $K_{D,Dimer}$ (Fig. S6). A rich source of information is NMR spectra, reflecting binding thermodynamics and kinetics[36]. We recorded $^1$H,$^{15}$N HSQC TROSY spectra at increasing P-domain concentrations (cf. Fig. 1a) but have not yet obtained a backbone assignment since all attempts to develop an efficient unfolding-refolding protocol for complete deuterium-hydrogen back exchange of backbone H$^N$ failed. In a previous NMR study of HuNoV P-dimers[33], we demonstrated that deuterium-hydrogen back exchange is crucial for the backbone assignment of norovirus P-domains. Yet, cross-peak intensities of unassigned dimer cross-peaks in a series of concentration-dependent $^1$H,$^{15}$N HSQC TROSY spectra of MNV P-domain can be used for an estimate of $K_{D,Dimer}$ yielding a value of ca. 31 μM (Fig. 1a, b, Fig. S6a, Table S2). To better define and quantify the monomer-dimer equilibrium, we turned to specific $^{13}$C side-chain methyl group labeling strategies[30,31,37] enabling the acquisition of methyl TROSY spectra. An advantage of $^{13}$C-methyl group labeling is the relative ease with which (partial) assignments of side-chain methyl group signals can be obtained, provided high-resolution crystal structure data for the protein in question are available. Methyl TROSY[38,39] experiments then allow detection of CSPs upon, e.g., ligand binding and provide experimental access to binding topologies as well as binding thermodynamics and kinetics[36,37].

**Assignment of $^{13}$C-methyl signals of the MNV apo P-domain.**
For the analysis of the P-domain monomer-dimer equilibrium, we synthesized MIL$^{proS}$VP$^{proS}$A labeled MNV CW1 P-domain. An assignment of $^{13}$C-methyl groups of CW1 P-dimers in the presence of saturating amounts of GCDCA had been previously completed[40]. The stabilization of MNV P-dimers upon binding of GCDCA will be addressed in much more detail below. However, to study P-domain dimerization an assignment in the absence of GCDCA was required. Unfortunately, methyl TROSY spectra of the apo-form of CW1 P-domains showed insufficient S/N to perform NMR experiments required for the assignment of methyl group resonances, preventing a "direct" assignment. Saturation of P-domains with GCDCA shifts the monomer-dimer equilibrium completely to the dimeric state, yielding methyl TROSY spectra with much better resolution and considerably improved S/N. A combination of 4D HMQC-NOESY-HMQC experiments, point mutations, and paramagnetic NMR experiments then furnished an assignment[40] (Fig. S7). An unexpected complication was the presence of cis and trans isomers at Pro361, leading to a larger number of cross-peaks than expected. At this point, we focus on the assignment of the methyl TROSY spectrum of the apo-form of the MNV P-domain starting with the assignment of the GCDCA-saturated form. Briefly, we followed peak intensities in methyl TROSY spectra with increasing concentrations of GCDCA to identify groups of peaks belonging to the same methyl group in apo P-domain monomers, apo P-dimers, and GCDCA-bound P-dimers. The cross-peak patterns of the apo-state of the protein and the GCDCA-saturated form differ dramatically. For many methyl TROSY cross-peaks Euclidian distances $\Delta\nu_{Eucl}$ between the apo and the GCDCA-bound form of P-domains are in the range of 50–150 Hz and correspond to slow exchange on the chemical shift time scale, leading to separate sets of peaks for the apo P-domain and GCDCA-bound P-dimers (Fig. S8a). Assignment followed a nearest-neighbor approach[36,41–43], which simply grouped peaks emerging upon removal of GCDCA to an assigned cross-peak into one group belonging to the same methyl group. Other cross-peaks in the intermediate exchange regime ($\Delta\nu_{Eucl} = 20$–$50$ Hz) and in the intermediate to fast exchange regime ($\Delta\nu_{Eucl} < 20$ Hz) were fewer but easier to assign (Fig. S8b-d). Using this approach, we assigned 58 apo P-domain cross-peaks. No CSPs were determined for three of these cross-peaks as apo-dimer peaks were not sufficiently resolved, e.g., the L232 cross-peaks (Fig. S15a). For 9 of the remaining 55 peaks, the corresponding monomer peaks can be unambiguously identified using the nearest-neighbor approach, allowing further analysis of the monomer-dimer equilibrium (Table S3 and Fig. S9).

Subjecting $^{13}$C-methyl (MIL$^{proS}$VP$^{proS}$A) labeled apo P-domain to methyl TROSY experiments at different protein concentrations allowed direct monitoring of monomer-dimer exchange (Fig. S9). The series of concentration-dependent methyl TROSY 2D spectra was then fitted using direct quantum mechanical simulations with the program TITAN[44,45]. Peak pairs with a monomer peak unambiguously allocated to a specific dimer peak were defined as regions of interest, and the parameters of a two-state exchange kinetic model were fitted to the experimental lineshapes[45]. The analysis provides the dissociation constant $K_{D,Dimer}$ of $6.9 \pm 0.6\,\mu M$ and the dissociation rate constants $k_{off,Dimer}$ of $1.25 \pm 0.21\,s^{-1}$ (Fig. S9 and Table S2).

**Dimer dissociation rate constants of murine and human norovirus P-domains differ by six orders of magnitude.** To put our results in a broader perspective we also studied the exchange between monomers and dimers for HuNoV P-domains. MNV and HuNoV P-dimers share a remarkably similar tertiary structure, including conservation of most secondary structure elements

(Fig. S10), although sequence identity on the amino acid level is poor (about 30%). We chose GII.4 Saga P-dimers as HuNoV representatives that have been studied by NMR in our laboratory before[33,46,47]. Native MS studies[48] had indicated traces of monomeric species for GII.4 Saga P-domains. On the other hand, NMR experiments did not lead to any signals resulting from monomers[33,47]. It follows that dimer dissociation must be fairly slow and, in turn, P-dimers must be quite stable. To obtain a dissociation rate constant for GII.4 Saga P-dimers we employed analytical ion-exchange chromatography (IEX), following a protocol established before to study spontaneous deamidation of a specific asparagine residue, Asn 373, located in the histo-blood group antigen (HBGA) binding pocket[33]. Briefly, deamidation exclusively furnishes an iso-aspartate residue (iD) in position 373 and essentially switches off HBGA binding. We took advantage of the charge differences between the stable, fully deamidated dimers (termed "iD/iD"), carrying two extra negative charges, and a non-deamidating GII.4 Saga P-domain point mutant N373Q ("Q/Q"). Mixing Q/Q and iD/iD P-dimers leads to an exchange of monomeric units over time yielding three types of P-dimers, homodimers iD/iD and Q/Q as well as mixed P-dimers iD/Q. The charge differences between iD/iD, Q/Q, and iD/Q allow separation using analytical IEX (Fig. 2a). Starting with a 1:1 mixture of pure iD/iD and Q/Q P-dimers and subjecting aliquots of the mixture to quantitative IEX analysis at fixed time intervals for 38 days yields the concentrations of homodimers and mixed dimers as a function of time (Fig. 2b). Assuming constant monomer concentrations throughout the experiment, the exchange of monomeric P-domains only depends on the rate constant $k_{off}$ for the dissociation of P-dimers[49]. Fitting the concentration of mixed dimers to an exponential function (Eq. 3) yields a dissociation rate constant $k_{off}$ of $1.5 \times 10^{-6}\,s^{-1}$, six orders of magnitude smaller than the corresponding value for MNV P-dimers. These results are supported by native MS experiments performed on GII.4 Saga N/N P-dimers. The mass spectra almost exclusively show signals from P-dimers (Fig. 2c).

**GCDCA stabilizes MNV P-domain dimers.** SEC elution profiles of samples of CW1 P-domains in the presence of saturating amounts of GCDCA are insensitive to protein concentration, and the SEC elution volume translates into a molecular weight of ca. 64 kDa as expected for P-domain dimers (Fig. 3a). Concurrently, the thermostability of CW1 P-domains improves considerably with increasing GCDCA concentrations (Fig. 3b and Fig. S11). Of note, the increased thermostability also shows on the level of complete virions as reflected by increased virus titers after incubation for several hours at elevated temperatures in the presence of GCDCA (Fig. S11). No such effects are observed in the presence of other non- or weakly binding bile acids such as cholic acid (CA), taurocholic acid (TCA), or taurochenodeoxycholic acid (TCDCA) (Fig. S12).

Measurement of longitudinal ($T_1$) and transverse ($T_{1\rho}$, $T_2$) $^{15}$N relaxation times yielded $T_1/T_2$ ratios reflecting rotational and internal motions on the ps–ns time scale[50]. The average values considerably increase upon the addition of GCDCA, indicating an increase in the molecular tumbling time and thus also in molecular weight (Fig. S13). These findings underscore that P-domain dimers are markedly stabilized by GCDCA. Finally, we note that removal of GCDCA yields unperturbed NMR spectra of the apo-form, showing that the process is fully reversible (Fig. S14).

**Chemical shift perturbation NMR experiments reveal long-range effects upon GCDCA binding.** To obtain more insight into GCDCA-assisted P-domain dimerization we performed

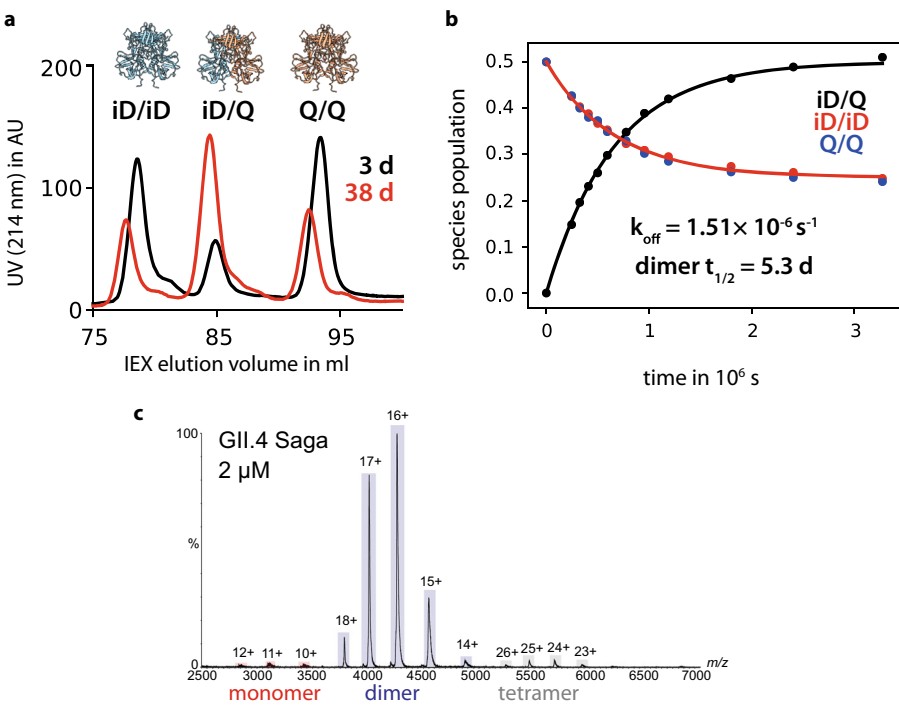

**Fig. 2 Dimer exchange kinetics of human NoV GII.4 Saga. a** P-domain samples which were deamidated at position 373 into a negatively charged iso-aspartate (iD) residue and point mutants with a neutral Gln at this position were mixed 1:1 and incubated at 298 K. The fraction of mixed 373iD/Q dimers was quantified by analytical ion-exchange chromatography at selected time intervals. Homo- and mixed dimer fractions reached the expected 1:2:1 ratio after ~40 d. **b** Curve fitting yields a $k_{off}$ of $1.5 \pm 0.02 \times 10^{-6}$ s$^{-1}$, corresponding to a dimer half-life time of 5.3 d. **c** Native mass spectra of human NoV GII.4 Saga P-domain (N/N P-dimers) confirm mostly dimeric stoichiometry even at low protein concentration.

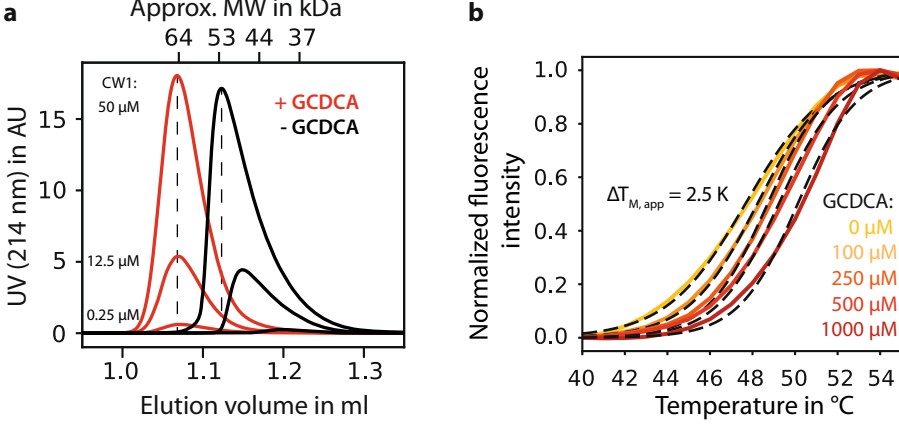

**Fig. 3 The MNV P-domain readily dimerizes in presence of GCDCA. a** Size-exclusion chromatography of the MNV CW1 P-domain in the presence (red curves) and absence (black curves) of GCDCA reveals a major change in elution behavior. The presence of saturating amounts of GCDCA (100 μM) shifts the apparent molecular weight of the P-domain to that of the dimer (64 kDa) regardless of the protein concentration used. Six representative chromatograms are shown with a protein concentration range of 250 nM–50 μM. The disappearance of the protein concentration-dependent elution behavior indicates a dramatic stabilization of the dimeric protein in presence of its ligand. **b** The thermal stability of the CW1 P-domain increases with increasing GCDCA concentrations. Apparent CW1 melting temperatures were obtained by differential scanning fluorimetry and curve fitting to a Boltzmann model (dashed lines).

chemical shift perturbation (CSP) NMR experiments. CSPs of $^{13}$C-methyl group signals of MIL$^{proS}$V$^{proS}$A labeled CW1 P-domain were monitored with increasing concentrations of GCDCA using methyl TROSY spectra (Fig. 4). Based on the assignment of $^{13}$C-methyl signals of the P-domain saturated with GCDCA[40] this titration also provided a partial assignment of cross-peaks belonging to the apo-form of the P-domain as described above. For the interpretation of CSPs reported as Euclidian distances $\Delta\nu_{Eucl}$[51] (Fig. S15) we first calculated a

minimum threshold for significant CSPs, following a protocol we had used before[33]. The calculation provides a threshold value of 5.1 Hz (Figs. S16, S17). CSPs larger than this value are significant with a confidence level of 99.99%. Significant CSPs not only map on the GCDCA binding pocket but also reflect long-range effects, indicating potential conformational changes at remote sites (Fig. 4a, b). We then classified CSPs above the threshold of 5.1 Hz as "small", values larger than 10 Hz and smaller than 20 Hz as "medium" and CSPs larger than 20 Hz as "large". "Medium" and

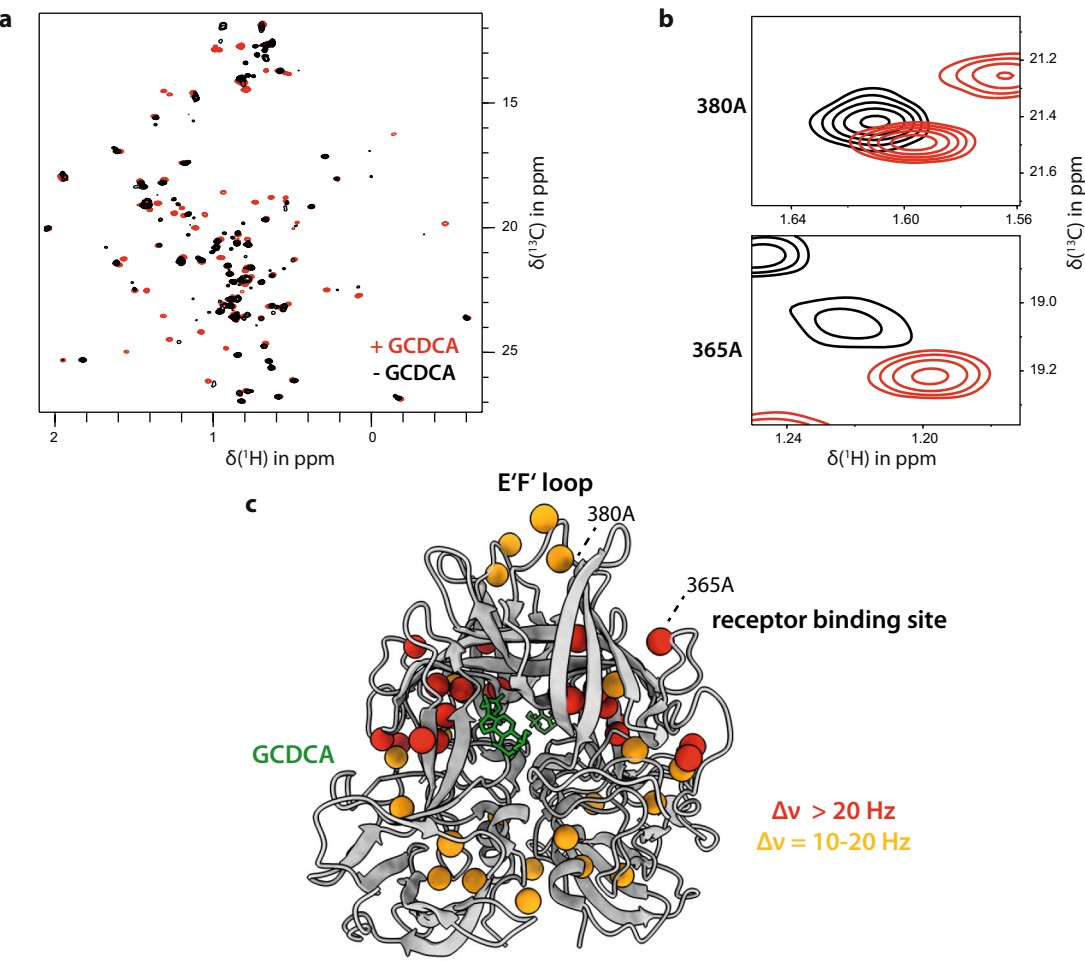

**Fig. 4 Chemical shift perturbations in MNV CW1 P-domains upon GCDCA binding. a** Methyl TROSY spectra of 50 μM $^{13}$C-methyl group-labeled (MILVA-labeled) MNV P-domains (black spectrum) and titrated with 550 μM GCDCA (red spectrum). **b** The spectra reveal CSPs ($\Delta\nu_{Eucl}$, see Table S3) also in regions up to 20 Å apart from the GCDCA binding pocket. **c** Medium (orange) and large (red) CSPs mapped onto the crystal structure model of the MNV P-domain (PDB 6E47). For the division of significant CSPs into "small", "medium", and "large" effects cf. Figs. S16, S17.

"large" CSPs were mapped on the crystal structure model (Fig. 4c). Signals experiencing large CSPs are characterized by separate peaks for the apo-form and the GCDCA-bound form, reflecting slow exchange on the chemical shift time scale. Not surprisingly, most of these signals correspond to methyl side chains of amino acids in proximity to the GCDCA binding pocket, e.g., A393 and M436 (Fig. 4c and Table S3). Twelve signals could not be traced back to the apo-form. Either the associated CSPs are outside a certain Euclidian distance, which must be larger than ca. 30 Hz or the missing signal is exchange broadened with the exchange reaction not necessarily being the exchange between apo and bound state. For further analysis we have only considered peaks where CSPs could be measured (Table S3).

Interestingly, some of the CSPs with $\Delta\nu_{Eucl}$ larger than 20 Hz are long-range effects. For example, A365 and I358 are in a loop region (D'E' loop) that is part of the receptor (CD300lf) binding site[11] and belong to this group of residues. Signals with CSPs in the range of 10–20 Hz correspond to intermediate exchange on the chemical shift time scale and map to amino acids across almost the entire protein, including regions that show no structural differences between bound and unbound states in crystal structure models (Fig. S18). Within this set of signals interesting long-range CSPs are observed in the E'F' loop that is recognized by neutralizing mAb A6.2, e.g., for A380 (Fig. 4).

**Long-range CSPs are correlated with an escape from neutralizing mAbs**. The observation of long-range CSPs prompts experiments that may allow correlations with biological effects. For instance, it has been shown that substitution of specific P-domain amino acids leads to escape from recognition by neutralizing mAbs[24]. As another example, it has been reported that GCDCA abrogates the binding of neutralizing mAbs to MNV P-domains in neutralization assays[22]. Here, we revisited some of the mAb-binding experiments to compare to the NMR results. We first investigated the effects of GCDCA and TCA on MNV infection in the presence and absence of neutralizing mAbs 2D3, 4F9, and A6.2. We find that the addition of GCDCA but not TCA blocks mAb recognition, allowing for viral infection even in the presence of neutralizing mAbs (Fig. S19). This result follows published data and correlates with long-range CSPs within the E'F' loop (Fig. 4), which is recognized by the mAbs.

We then tested whether the presence of GCDCA would affect protection by the neutralizing mAb A6.2 against infection of cells with MNV virions, carrying the escape mutations V378F or L386F[24]. The mutations are flanking the E'F' loop whose orientation is critical for mAb recognition and essentially abrogate binding to mAb A6.2 (Fig. S20). The addition of GCDCA has no effect on infection showing that the addition of GCDCA does not make a difference for the escape mutants. In any case, no binding of mAb A6.2 is observed.

**A model for MNV P-domain dimerization and GCDCA binding**. CSPs upon ligand binding are usually larger in $^1$H,$^{15}$N TROSY HSQC spectra than in methyl TROSY spectra. This is also observed for the binding of GCDCA to MNV P-dimers. Although a backbone assignment is not available the effect of adding GCDCA on monomer vs. dimer peaks can be better evaluated from cross-peaks in $^1$H,$^{15}$N TROSY HSQC spectra. Peaks that were unambiguously assigned to either the monomeric or the dimeric state of the apo P-domain show different behavior upon titration with GCDCA. In general, monomer cross-peaks disappear and show no CSPs. In contrast, apo-dimer cross-peaks show either CSPs typical for intermediate-to-fast exchange on the chemical shift time scale or the appearance of new cross-peaks with different resonance frequencies, indicating slow exchange (Fig. S21). This observation suggests that P-domain monomers cannot bind to GCDCA and that P-domain dimerization is required for GCDCA binding. This is consistent with the crystal structures where each of the two symmetrical GCDCA binding pockets are comprised of amino acids from both monomeric units (PDB 6E47)[11]. Accordingly, increasing GCDCA concentrations lead to depletion of monomers accompanied by a shift of the overall equilibrium towards the dimeric, bound state. This is supported by the disappearance of all monomer signals at equimolar concentrations of GCDCA, accounting for most of the observed complex spectral changes. Thus, for the following, we assume that P-domain dimer formation must precede GCDCA binding.

The appearance of NMR resonances from a dimeric molecule across a titration with a ligand can provide insight into cooperativity and intersubunit communication[52]. In our case, no additional free or bound resonances were observed that might be associated with structural changes upon ligand binding at a distal site. The simplest binding model that accounts for these observations contains three coupled equilibria, with a single dissociation constant for ligand binding ($K_{D,GCDCA}$, indicating that the two symmetrical GCDCA binding pockets of P-dimers are equal and independent) and free/bound chemical shifts that do not depend on the occupancy of the distal binding site (Fig. 5a). For the first equilibrium, the monomer-dimer exchange, we have derived a dissociation constant $K_{D,Dimer}$ of 7 μM from independent ab initio analysis of methyl TROSY cross-peaks of exchanging monomers and dimers as described above (Fig. S9). These three states (P-domain monomers, P-dimers, and P-dimers with GCDCA bound) were at least partially resolved for 11 methyl resonances. We implemented this kinetic model within the TITAN lineshape analysis package and fitted the model to the set of experimental signals (Fig. 5b and Fig. S22). Within this model, GCDCA binding is characterized by a $K_{D,GCDCA}$ of $10.5 \pm 0.5$ μM and a dissociation rate constant $k_{off,GCDCA}$ of $25.8 \pm 3.5$ s$^{-1}$.

Alternatively, the absence of signals corresponding to the asymmetric, single-bound P-dimer state could also indicate strong positive cooperativity between the first and second binding events. In this model, the single-bound state would be sparsely populated and essentially unobservable. To test whether our data would be consistent with this binding model, we also implemented this model into TITAN. Specifically, we introduced two parameters α and β representing thermodynamic and kinetic cooperativity respectively, as defined in Eq. 1:

$$K_{D2,GCDCA} = K_{D1,GCDCA} \cdot 10^\alpha \text{ and } k_{off2,GCDCA} = k_{off1,GCDCA} \cdot 10^\beta \tag{1}$$

where $K_{D1,GCDCA}$ and $K_{D2,GCDCA}$ are the dissociation constants for the single and double-bound species, respectively, and $k_{off1,GCDCA}$ and $k_{off2,GCDCA}$ represent the rate constants for the

dissociation of the first and the second GCDCA ligand. Since no information on the chemical shifts of the postulated hidden state is available, we positioned the signals corresponding to those hidden states randomly into noise regions of the spectrum for each set of cross-peaks analyzed. Allowing α, β, $K_{D1,GCDCA}$ and $k_{off1,GCDCA}$ to vary during lineshape fitting yielded a value of zero for β, indicating no apparent difference in dissociation rates. Therefore, we reduced the number of parameters in subsequent fits by fixing β to zero, leading to a single dissociation rate constant, $k_{off}$. Lineshape fitting and bootstrap error analysis[45] yielded values of $K_{D1,GCDCA} = 3 \pm 2$ mM, $k_{off} = 14 \pm 32$ s$^{-1}$, and $\alpha = -4.7 \pm 0.6$, corresponding to $K_{D2,GCDCA}$ approximately equal to 60 nM (Fig. 5a). While data fit well to this model (Fig. 5b), the quality of the fit is not noticeably improved relative to the simple non-cooperative model analyzed above. Moreover, parameter uncertainties are large, in the order of the values themselves, with strong covariance between parameters, indicating that the fitting problem is effectively underdetermined. Therefore, while we cannot exclude a strongly cooperative model, based on the current data no reasons exist to prefer this over the simpler, non-cooperative binding model.

## Discussion

In the complete viral capsid, P-domains are connected to the underlying shell (S-domains) via an extended hinge comprising several amino acids. The possibility of loose instead of tight interactions between P-domains arranged as dimers in the viral capsid has not been considered before. Crystal structures of the P-domains feature dimers, insinuating that these dimers are "stable". On the other hand, native MS data on HuNoV P-domains suggest that monomer-dimer equilibria play a role[48], and for MNV P-domains it has been noted that P-dimers may dissociate into monomers in solution[11]. These observations prompted a more detailed analysis of norovirus P-dimer dissociation in terms of thermodynamic stabilities and dissociation rates. We have used protein NMR spectroscopy complemented by other biophysical techniques such as native MS, analytical IEX, and SEC/SEC-MALS to obtain a comprehensive picture of P-dimer dissociation, revealing notable differences between human and murine noroviruses. Importantly, MNV P-domain dimer formation, in contrast to HuNoV, is strongly promoted by GCDCA binding, at the same time associated with loop reorientations and subsequent immune escape.

A key finding of this study is that MNV P-domains co-exist as a mixture of dimers and monomers in an aqueous solution under acidic conditions, which corresponds to the physiological environment in the gastrointestinal tract of the host[53]. The presence of a P-domain monomer-dimer equilibrium has been independently shown by NMR, native MS, and analytical size-exclusion chromatography (Fig. 1). The quantitative analysis of P-dimer dissociation has been based on ab initio simulations[54] of concentration-dependent methyl TROSY spectra of MNV.CW1 P-domains using the program TITAN[44,45], yielding a dissociation constant $K_{D,Dimer}$ of 6.9 μM, and corresponding on- and off-rate constants $k_{on,Dimer}$ and $k_{off,Dimer}$ of $1.4 \times 10^5$ M$^{-1}$ s$^{-1}$ and 1.2 s$^{-1}$, respectively (Table S2). The fast dissociation of MNV P-dimers contrasts with the slow dissociation of human norovirus P-dimers. The dissociation rate constant of P-dimers of a prevalent human norovirus strain (GII.4 Saga) is about six orders of magnitude smaller than for MNV P-dimers (Fig. 2), indicating a dissociation constant in the picomolar range. In addition, we have shown previously that GII HuNoV P-dimers provide no high-affinity binding pockets for GCDCA. Rather, GCDCA binds to P-dimers with dissociation constants in the millimolar range[46]. In the case of the GII.4 Saga strain, we have shown that the

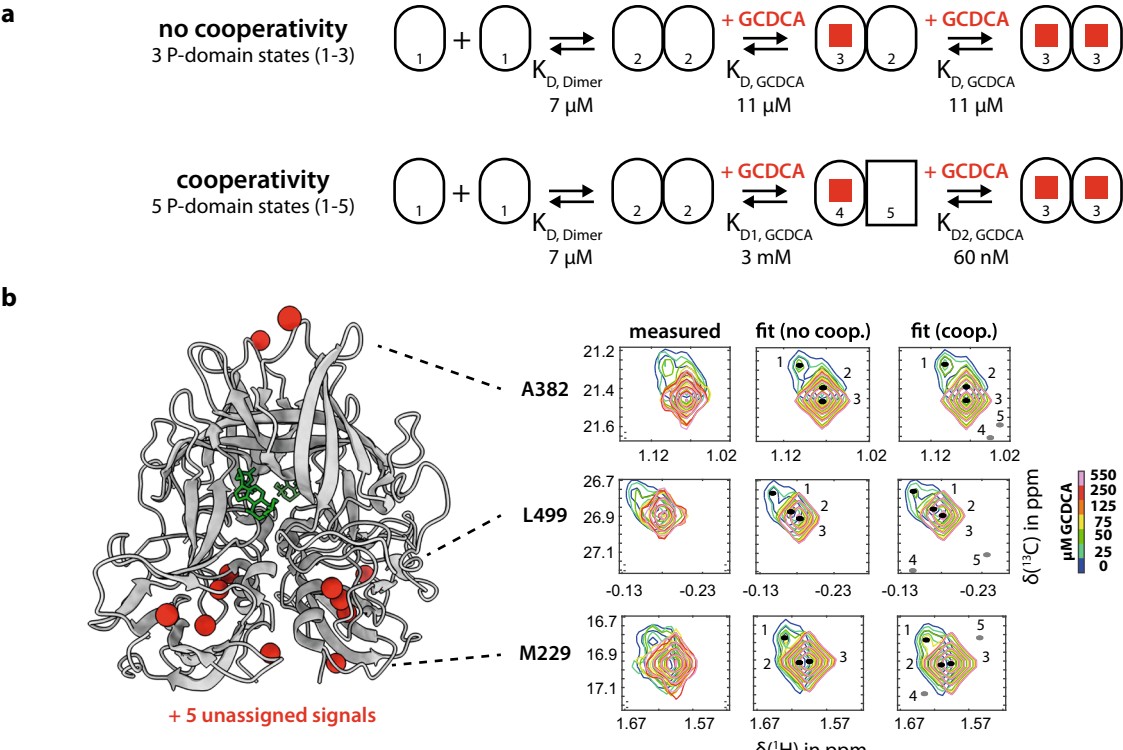

**Fig. 5 Possible GCDCA binding models. a** Methyl TROSY-based GCDCA titration data (Fig. 4) match two binding models: a 3-state binding model in which dimerization is followed by two consecutive, independent binding events with a dissociation constant $K_D$ of 10.5 μM and a model with strong cooperativity in which binding of the first GCDCA molecule dramatically increases the affinity for the second. In both models, the monomeric protein (1) is binding-incompetent and is depleted at higher ligand concentrations by removal of the unbound dimer (2) through subsequent binding reactions. **b** Using TITAN, both binding models were fitted to titration data where cross-peaks of monomeric (1), dimeric (2), and ligand-bound states (3) were at least partially resolved. Cross-peaks of A382, L499, and M229 are shown as examples (for the complete set of cross-peaks see Fig. S22). Note, that strong cooperativity inherently implies the existence of two unobservable P-domain states (4) and (5) in the single-bound dimer. Signal positions corresponding to these states were positioned randomly during the fit procedure. The dissociation constant $K_{D,Dimer}$ was determined independently by TITAN analysis of a P-domain concentration series (Fig. S9).

respective binding site is located at the bottom of the P-dimers close to the C-terminus[46] and not at the dimer interface. For a rare genotype, GII.1, yet another bile acid-binding site has been observed underneath the HBGA binding site[55]. It appears that GCDCA promotes murine and human noroviruses infection via quite different modes of action.

Results from cryo-EM studies of MNV capsids and of MNV P-domains have led to an overall picture where GCDCA binding leads to compression of virus particles, priming the virus for binding to the CD300lf receptor and at the same time escaping antibody-mediated neutralization[22]. Our results expand this picture, suggesting that the dimeric arrangement of capsid proteins in MNV is less static than thought. It is unlikely that P-domains completely dissociate into monomers when embedded in the virus capsid. However, the much lower stability of isolated MNV P-dimers as compared to human P-dimers should reflect fewer inter-domain contacts. To test this we used the PRODIGY server[56,57] for the prediction of dissociation constants of P-dimers using crystal structure data for prototype murine (PDB 3LQ6[21]) and human (PDB 4OOX[58]) P-dimers (Table S6). The dimer-interface analysis yields much fewer inter-domain van der Waals contacts for MNV P-dimers (PDB 3LQ6) than for human GII.4 Saga P-dimers (4OOX), resulting in a predicted difference of binding affinities of $\Delta\Delta G$ of 4.2 kcal/mol, which translates into dissociation constants $K_D$ differing by almost four orders of magnitude. As expected, this estimate is not a perfect match, but it nicely rationalizes the experimental data (Table S2). We

conclude that the reduced number of attractive forces in an MNV virus capsid allows relative movement of P-domains within the so-called A/B and C/C dimers[59]. This corresponds well with the observation that the formation of contracted virus particles upon GCDCA binding requires rotation of monomers in the A/B P-dimers to prevent steric clashes between the P- and the S-domain[22]. It is tempting to hypothesize that this rotating motion of P-dimers as part of the viral capsid and dissociation of isolated P-dimers in solution is linked to the same molecular origin. Following this argument, the solution conformation of P-dimers with GCDCA bound likely will match the conformation of P-dimers in contracted MNV capsids.

The NMR data presented provide a more detailed picture of GCDCA binding to MNV P-domains. Affinity data based on isothermal titration calorimetry (ITC) experiments had been published, quoting a dissociation constant $K_D$ of 6 μM[11]. This matches the dissociation constant $K_{D,GCDCA}$ of 11 μM (Table S2, Fig. 5) derived from NMR experiments quite well. However, our NMR study shows that the formation of MNV P-dimers complexed with GCDCA involves more than the attachment of bile acid ligands to a dimeric protein. Experimentally, three conformational states (states "1", "2", and "3" in Fig. 5a) can be distinguished for each P-domain, leading to the simplest binding model where dimerization of P-domains precedes GCDCA binding. The GCDCA-bound P-dimers are in a different conformational state (state "3") as compared to the apo P-dimers (state "2"). In other words, the binding of GCDCA is associated

with conformational transitions. We did not find any evidence supporting a cooperative binding model involving five conformational states (Fig. 5a). We conclude that the simple three-state binding model (Table S2, Fig. 5a) is adequate to describe dimerization and GCDCA binding. Of note, dissociation of GCDCA from the P-dimer (state "3") is an order of magnitude faster than dissociation of apo P-dimers (state "2") into monomers (state "1").

As already mentioned, stabilization of MNV P-dimers by binding to GCDCA has parallels with the effect of GCDCA on the overall shape of MNV capsids. We extend this picture by the observation from MNV infection assays that GCDCA but not TCA promotes infection in the presence of neutralizing mAbs (Fig. S19). This observation matches well with the finding that GCDCA but not TCA stabilizes MNV P-dimers (Fig. S12). It also supports the idea that overall structural changes of MNV capsids[22] are linked to the stabilization of P-dimers.

Long-range CSPs (Fig. 4) correlate with conformational changes of the respective protein regions upon GCDCA binding[51,60]. Therefore, CSPs for the methyl groups of alanine residues 380, 381, and 382 located in the E'F' loop (Fig. 4c and Fig. S18) may reflect the loop reorientation, which obstructs the binding of neutralizing antibodies as observed in cryo-EM and crystal structures[14,15,22]. All these positions are about 20 Å away from the binding site. It has been described that the transition between the "open" and the "closed" conformation is associated with the reorientation of the A'B', E'F', and the C'D' surface loops[21] in a concerted fashion. When GCDCA binds to the P-dimer, the C'D' loop moves up, pushing the E'F' and the A'B' loop into a conformation that is presumably not recognized by neutralizing antibodies[15,24]. In solution, the "open" and "closed" conformations likely exist as an equilibrium mixture in the absence of GCDCA. Upon addition of GCDCA this equilibrium is pulled to the "closed" state, which would account for the CSPs observed in the E'F' loop. The A'B' loop also undergoes conformational reorganization upon GCDCA binding but is lacking [13]C-methyl, reporter groups. Future work, e.g., based on measurements of residual [13]C–[1]H dipolar couplings[61,62] may complete the overall picture. Further insight is obtained from analyzing CSPs, not as Euclidian distances between peak positions but using the individual chemical shift changes in the [1]H and [13]C dimensions. Changes of [13]C chemical shifts of Ala methyl groups directly report on the backbone conformation[63,64], whereas Ile, Leu, Val, and Met methyl [13]C chemical shifts carry information on the side-chain rotamer populations[65–69]. Separate [1]H and [13]C chemical shift perturbations are summarized in Table S3. It is interesting to note that for the Ala residues in the E'F' loop it is almost exclusively the [13]C CSPs that contribute to the overall effect, confirming a conformational change of this loop upon GCDCA binding.

Other long-range effects are observed for amino acids belonging to the receptor (CD300lf) binding site that mainly involves the D'E' loop. For Ala365 and Ile358 CSPs with $\Delta\nu_{Eucl}$ of ca. 50 and 30 Hz, respectively, are observed (Table S3). Ala365 and Ile358 are close to Pro361, which exists in two configurations causing two methyl TROSY cross-peaks for each amino acid. Inspection of the CSP data shows that the [13]C chemical shifts are contributing most to the overall CSPs. For Ile358 this means that the side-chain conformation is affected[67], which is found in contact with the CD300lf receptor in the crystal structure[11]. In the light of these data, it may be speculated that GCDCA-binding modulates the corresponding loop region such that receptor binding is facilitated, consistent with the observation that GCDCA binding slightly improves the binding affinity for CD300lf[11]. As seen in Fig. 4 there are many long-range CSPs at the bottom of the P-domain, where the shell domain is linked to full-length VP1 capsid protein. These long-range effects may reflect subtle changes in conformational equilibria, which may facilitate the collapse of P-domains onto the S-domains.

Our study shows that the interaction of human and murine norovirus P-domains with GCDCA strikingly differs although in both species the presence of GCDCA promotes infection in cell culture experiments. To date, no human norovirus P-domain has been found with binding affinities for GCDCA or any other bile acid comparable to MNV. The observation that GCDCA binding stabilizes MNV P-dimers and that the dissociation rate constant $k_{on,Dimer}$ for HuNoV P-dimers (GII.4 Saga) is six orders of magnitude smaller than found for the apo MNV P-dimers (Table S2) prompts for more systematic studies into the stability of P-dimers of other noroviruses and eventually also of other caliciviruses.

## Materials and methods

**Protein biosynthesis.** P-domain proteins were synthesized according to a previously published protocol[33]. Briefly, *E. coli* BL21 DE3 cells were transformed with a plasmid containing genes for ampicillin resistance and a fusion protein of maltose-binding protein (MBP) and the P-domain, separated by two His8-tags and an HRV3C cleavage site. Unlabeled protein was expressed in terrific broth medium, whereas $[U-^2H,^{15}N]$ labeled P-domain was expressed in D2O-based minimal medium with deuterated glucose and [15]NH4Cl (Deutero) as sole carbon and nitrogen sources, respectively. For the synthesis of $[U-^2H,^{15}N]$-labeled protein, NaCl, KH2PO4, Na2HPO4, deuterated glucose, and [15]NH4Cl were dissolved in D2O (Deutero) and lyophilized before use. Fifty milliliters of LB medium were inoculated, and cells were grown overnight at 37 °C. A fraction of the medium corresponding to an OD600 of 0.05 in 40 mL was centrifuged and the bacterial pellet was resuspended in a minimal medium. Cells were grown until an OD600 of 0.4 was reached and then diluted to the final culture volume (0.5–1 L). At an OD600 of 0.8, expression was induced with 1 mM IPTG, and growth was continued at 16 °C until the stationary phase was reached. For the synthesis of $[U-^2H]$ [13]C-methyl labeled proteins, the isotopically labeled precursors ε-[13]C,[1]H3]-Met, δ1-[13]C,[1]H3]-Ile, γ2-[13]C,[1]H3]-Val, δ2-[13]C,[1]H3]-Leu, β-[13]C,[1]H3]-Ala labeled (MIL^proS^V^proS^A-labeling) were used as compiled in Table S5. The precursors were dissolved in a minimal medium (one-fifth of the final culture volume) and added to the culture when an OD600 of 0.8 was reached. Cell growth was continued at 16 °C. After 1 h, expression was induced by the addition of 0.1 mM IPTG, and growth was continued until the stationary phase was reached.

Cells were lysed using a high-pressure homogenizer (Thermo). The lysate was clarified by ultra-centrifugation and subjected to Ni-NTA affinity chromatography to yield the pure fusion protein. The fusion protein was cleaved by the addition of HRV3C protease and simultaneous dialysis against 20 mM sodium acetate buffer, 100 mM NaCl (pH 5.3). Ni-NTA affinity chromatography was repeated to separate the cleavage products. The P-domain protein was concentrated and applied to a preparative Superdex 16/600 200 pg size-exclusion column (GE) using the buffer as a running buffer. Amino acid sequences of the cleaved P-domains are given in Tab. S1 of the supplement. Protein concentrations were quantified by UV spectroscopy using a molar extinction coefficient of 46870 M[−1]cm[−1].

**Analytical size-exclusion chromatography.** Analytical size-exclusion chromatography has been performed with a Superdex 75 Increase 3.2/300 column (GE Life Sciences) and 20 mM sodium acetate, 100 mM NaCl (pH 5.3) as running buffer with a flow rate of 0.075 ml min[−1] at 278 K. To study the effect of ligand binding, 100 μM glycochenodeoxycholic acid (GCDCA, Sigma–Aldrich) was added to the running buffer. Protein samples were applied with a 10 μl sample loop using a complete loop filling technique. UV absorption was monitored at 280 nm and 214 nm simultaneously. A molecular weight calibration was performed with a mixture of 1.5 mg mL[−1] conalbumin (Sigma), 4 mg mL[−1] ovalbumin (Sigma), 1.5 mg mL[−1] carbonic anhydrase (Sigma), 4 mg mL[−1] cytochrome c (Sigma), and 1 mg mL[−1] aprotinin (Roth). Apparent molecular weights as a function of feed protein concentration were fitted to the equation below[35] to yield an apparent dissociation constant $K_D$ of the protein-protein interaction as well as apparent molecular weights of the pure monomer and dimer. To account for the dilution factor between sample application and final elution (10×), fitting was repeated using the respective lower protein concentrations to obtain a lower $K_D$ limit using Eq. 2:

$$MW_{obs} = MW_D + \frac{\sqrt{K_D(K_D + 8P_t)} - K_D}{4P_t}(MW_M - MW_D) \qquad (2)$$

with $MW_{obs}$ is the observed molecular weight according to the calibration of the column using standard proteins, $MW_D$ and $MW_M$ are the apparent molecular weights of the monomers and dimers, respectively, and $P_t$ is the total protein concentration applied onto the column.

**Hydrophobic interaction chromatography**. A 1 ml HiTrap Butyl-HP column (GE) was used for hydrophobic interaction chromatography. Protein samples (240 and 270 µg ml$^{-1}$ for pH and GCDCA dataset, respectively) were prepared in different buffers and then subjected to isothermal denaturation at 45 °C. The effect of GCDCA binding was studied using 20–1000 µM GCDCA in 20 mM sodium acetate buffer, 100 mM NaCl (pH 5.3) with 30 min incubation time, while experiments on pH-dependence were performed in 75 mM sodium phosphate buffer, 100 mM NaCl with pH values ranging from 5.7 to 8 with 10 min incubation time. Before loading to the column, 750 mM ammonium sulfate was added from highly concentrated stock. The protein bound to the column in 20 mM sodium acetate, 750 mM ammonium sulfate (pH 5.3), and was eluted using a linear gradient over 5 column volumes up to 100% of 20 mM sodium acetate (pH 5.3). A flow rate of 3 ml min$^{-1}$ was used. The UV integral at 214 nm of a non-heat-treated control sample was normalized to 1. Experiments were performed as duplicates.

**Exchange kinetics of human protruding domain dimers**. GII.4 Saga P-domain proteins were synthesized as described elsewhere[33]. We recently discovered an irreversible post-translational deamidation of Asn 373 into an iso-aspartate residue (iD). This reaction introduces a new negative charge into each monomer. The point mutant Asn373Gln does not undergo deamidation and carries no negative charge at this position. This charge difference between stable 373iD/iD and 373Q/Q homodimers can be exploited to analyze the kinetics of monomer exchange using analytical cation exchange chromatography (IEX) (Fig. 2). Both types of dimers were mixed 1:1 with a total protein concentration of 1.24 mg ml$^{-1}$ in 75 mM sodium phosphate buffer, 100 M NaCl, 0.02% NaN$_3$ (pH 7.3), and incubated at 298 K. At selected time points spanning 38 days aliquots were subjected to IEX experiments using the experimental conditions described in ref.[33]. With prolonged incubation times a new protein species can be observed corresponding to mixed dimers with an intermediate net charge. Exchange observed in such a mixing experiment only depends on $k_{off}$ as monomer concentrations are almost constant throughout the experiment[49].

$$f(iD/Q) = \frac{1}{2}(1 - e^{-k_{off}*t}) \qquad (3)$$

Equation 3 has been fitted to the fraction of mixed dimers $f(iD/Q)$ observed at different time points $t$ to yield:

$$k_{off} = 1.51 \pm 0.02 \times 10^{-6} s^{-1} \qquad (4)$$

The extrapolation of these results to native GII.4 P-domains relies on the assumption that the type of the amino acid at position 373 does not change the dimerization equilibrium. At this point, we would like to note that native MS experiments reproducibly reflect the presence of monomeric species for GII.4 Saga iD/iD P-dimers but not for the native, non-deamidated N/N P-dimers[48]. This is at odds with the NMR data, not indicating different dimer stabilities of the two forms of the dimers. At present we have no conclusive explanation for these diverging observations, but we believe the very different experimental setup used in MS and NMR is responsible, suggesting that MS and NMR shed light on different, may be related phenomena.

**Thermal shift assay**. CW1 P-domains (3 µM) were mixed with SYPRO orange (2×) fluorescent dye (5000× stock solution, Sigma) in 20 mM sodium acetate buffer, 100 mM NaCl, pH 5.3. The total reaction volume was 30 µl. Thermal shift assays were performed in a StepOne RT PCR system (Thermo Fisher) in a temperature range of 25–95 °C in 1 °C increment with a ramping rate of 100%, using the ROX filter. All experiments were performed as triplicates. Fluorescence intensity data were normalized and fitted to a Boltzmann model (Eq. 5) to yield apparent melting temperatures $T_M$ with $a$ governing the steepness of the transition[70].

$$f_U = \frac{1}{1 + e^{\frac{T_M - T}{a}}} \qquad (5)$$

**SEC-MALS**. One hundred microliters of CW1 protein samples were applied to a Superdex In 75 10/300 GL size-exclusion column (Cytiva) at concentrations between 10 and 200 µM. Size-exclusion chromatography was performed with a 1260 Infinity II HPLC system (Agilent) and a miniDawn light scattering detector (Wyatt) at a flow rate of 0.8 ml/min in 20 mM sodium acetate buffer, 100 mM NaCl, pH 5.3. Protein elution was detected by UV absorption at 280 nm and changes in the refractive index.

**NMR spectroscopy**. NMR spectra were acquired at 298 K on either a 500 MHz Bruker Avance III or a 600 MHz Bruker Avance III HD NMR spectrometer with TCI cryogenic probes. Samples of [$U$-$^2$H,$^{15}$N]-labeled proteins were prepared in 20 mM sodium acetate buffer, 100 mM NaCl, 500 µM DSS-d$_6$, and 10% D$_2$O (pH* 5.3) unless stated otherwise. Methyl group-labeled proteins were prepared in D$_2$O containing 20 mM sodium acetate-d$_3$, 100 mM NaCl, and 100 µM DSS-d$_6$, (pH* 5.26). pH values of samples with varying amounts of D$_2$O are given as uncorrected pH-meter readings pH*.

NMR spectra were processed with TopSpin v3.6 (Bruker), signal intensities were obtained with CcpNmr Analysis v2.4.2[71]. Acquisition parameters are given in Table S4 or in the respective figure legends. Protein concentrations in NMR experiments concerning P-domain dimerization were 12.5–226 µM in $^1$H,$^{13}$C HMQC-based experiments with methyl group-labeled samples (600 MHz), and 24–200 µM in $^1$H,$^{15}$N TROSY HSQC-based experiments (500 MHz). For GCDCA titration experiments, 100 µM [$U$-$^2$H,$^{15}$N]-labeled P-domain (500 MHz) and 50 µM MIL$^{proS}$VProS A -labeled protein (600 MHz) were titrated separately with GCDCA from highly concentrated ligand stock solutions prepared in the respective sample buffers up to final concentrations of 300 and 550 µM, respectively.

Titrations of GII.4 NoV P-domains with HBGAs were used to obtain Euclidean CSPs measured as Euclidian distances $\Delta\nu_{Eucl}$ according to Eq. 6:

$$\Delta\nu = \sqrt{\Delta\nu_H^2 + \Delta\nu_C^2} \qquad (6)$$

with $\Delta\nu_H$ and $\Delta\nu_C$ being the CSPs in the respective dimension in Hz.

$^{15}$N backbone relaxation data were obtained at 600 MHz using TROSY-based pulse schemes for measurement of $T_1$ and $T_{1\rho}$ relaxation times[50]. Spectra were acquired with 128 increments in the indirect dimension and a relaxation delay of 3 s. Delays in the pulse sequence were 0 s, 0.36 s, 0.6 s, 1 s, and 2.48 s for the determination of $T_1$ and 1 ms, 15 ms, 30 ms, and 50 ms for the determination of $T_{1\rho}$. Both experiments contained a spin-lock temperature compensation element of up to 50 ms. The spin-lock field strength $\omega$ in the $T_{1\rho}$ experiment was 1.4 kHz, the carrier frequency in the $^{15}$N dimension was 117.5 ppm.

**Global lineshape analysis of methyl TROSY spectra**. Lineshape analysis of cross-peaks of $^1$H,$^{13}$C HMQC spectra of MIL$^{proS}$VProS A -labeled samples employed the program TITAN[45] was performed with a different protein concentration as stated above using TITAN's built-in dimerization model. Spectra were processed in NMRPipe[72] before analysis. The shell scripts used are compiled in Supplementary Note 1. First, positions and linewidths of thirteen isolated putative monomer peaks were fitted using spectra of 12.5 µM and 25 µM P-domain. Then, positions and linewidths of the corresponding dimer peaks in spectra with 226 µM and 100 µM P-domain have fitted accordingly. Finally, peak positions of monomer and dimer peaks were held constant and all linewidths, the dissociation rate k$_{off}$, and K$_{D1}$ were fitted using the dimerization binding model at all six P-domain concentrations with the linewidths for monomer and dimer signals derived above as starting values. Parameter uncertainties were obtained by bootstrap error analysis.

TITAN lineshape analysis of $^1$H,$^{13}$C HMQC spectra resulting from a titration of GCDCA to a sample of MIL$^{proS}$VProS A -labeled MNV CW1 P-domain required implementation of new binding models into TITAN as described in the results part. The TITAN software including the extensions is freely available from https://www.nmr-titan.com. For the titration, a series of $^1$H,$^{13}$C HMQC spectra of MIL$^{proS}$VProS A -labeled P-domain with increasing GCDCA concentrations was acquired and processed with NMRPipe. Linewidths and chemical shifts of monomers (state 1) and dimers (state 2) were fitted using the apo spectrum. Linewidths of the fully bound P-domain dimer (state 3) were fitted using the spectrum of the P-domain in presence of 550 µM GCDCA. For models including cooperativity, chemical shifts of states 4 and 5 were set to arbitrary positions as described in the results part. Linewidths, off-rate constants k$_{off}$, dissociation constants K$_D$, and cooperativity parameters α and β (cf. Eq. 1) were then fitted using spectra at all seven ligand concentrations and employing previously determined linewidths as starting values.

**Mass spectrometry (MS)**. Proteins were subjected to buffer exchange to 150 mM ammonium acetate, pH 5.3 (murine P-domains), or pH 7.3 (GII.4 Saga P-domains) at 4 °C via Micro Bio-Spin 6 columns (Bio-Rad) according to the manufacturer's protocol. Native MS measurements were performed using 2–92 µM purified P-domains. Mass spectra were acquired at room temperature (25 °C) in positive ion mode on an LCT mass spectrometer modified for high mass (Waters, UK and MS Vision, the Netherlands) with a nano-electrospray ionization source. Gold-coated electrospray capillaries were produced in-house for direct sample infusion. Capillary and sample cone voltages were 1.3 kV and 200 V, respectively, for MNV07, CR10, and Saga P-domains, and 1.5 kV and 324 V for CW1 P-domains. The pusher was set to 100–150 µs. Pressures were 7 mbar in the source region and $6.2 \times 10^{-2}$ to $6.5 \times 10^{-2}$ mbar argon in the hexapole region. A spectrum of a 25 mg/mL cesium iodide solution from the same day was applied for calibration of raw data using the MassLynx software (Waters, UK).

**Determination of dissociation constants from MS**. Monomer and dimer peak areas were summed overall corresponding charge states. The relative dimer peak area was calculated and plotted against the total protein concentration ($P_t$). The $K_D$ was then determined by global non-linear least-squares fitting of Eq. 7[73] to the dataset using OriginPro 2016 software:

$$\frac{[D]}{P_t} = 1 - \frac{\sqrt{\frac{K_D}{2}\left(P_t + \frac{K_D}{8}\right)} - \frac{K_D}{4}}{P_t} \qquad (7)$$

**Alignment of crystal structure models**. We used PyMOL[20] and ChimeraX[74,75] for aligning crystal structure models. For aligning the Cα atoms of the P-domains in PDB 6e47 and PDB 6c6q we used PyMOL by applying the following sequence of commands:

```
fetch 6c6q 6e47, async = 0

align 6e47&c.A&n.ca, 6c6q&c.B&n.ca, object = Pdimer_calpha
```

**Cell culture**. Murine microglial cells (BV-2) were maintained in Dulbecco's Modified Eagle Medium (Gibco) supplemented with 5% FCS (C-C-Pro), 2 mM L-glutamine (Biozym), 0.1 mM non-essential amino acids (Biozym), and 100 units/ml penicillin and streptomycin (Biozym) (DMEM-5) as described[76]. Cells were incubated at 37 °C with 5% CO$_2$ and 95% humidity. Hybridoma suspension cell lines producing 2D3, 4F9[77], and A6.2[78] antibodies were a kind gift from Christiane Wobus (University of Michigan, USA). Briefly, cells were maintained in Iscove's Modified Dulbecco's Medium (IMDM, Life Technologies) supplemented with 10% FCS (C-C-Pro), 0.5 mM L-glutamine (Biozym), and 100 units/ml penicillin and streptomycin (Biozym) (IMDM-10). Cells were cultivated in spinner flasks at 30 rpm at 37 °C with 5% CO$_2$ and 95% humidity with loosened lids for gas exchange. Passaging was performed by replacing at least 50% of the 'old' conditioned medium with 50% fresh medium diluting the cells to a concentration of ca. 1 ×10$^5$ cells per ml.

**Purification of antibodies**. Antibodies 2D3, 4F9, and A6.2 were produced from respective hybridoma suspension cultures. Briefly, cultures were grown to a density of at least 1 × 10$^6$ cells per ml. A minimum of 10$^9$ cells were pelleted by centrifugation (10 min 1000 rpm) and resuspended in serum-free medium (Capricorn Scientific Hybridoma Plus) supplemented with 100 units/ml penicillin and streptomycin (Biozym) adjusted to a density of 1 × 10$^6$ cells per ml. Cultures were incubated in spinner flasks at 37 °C with 5% CO$_2$ for 3 days. Suspensions were clear-centrifuged (30 min 3000 rpm 4 °C) and the cell pellet was discarded. The supernatant containing the antibodies was sterile filtered (0.2 μM) and antibodies were precipitated with a 45% saturated solution of ammonium sulfate (final concentration). IgA Antibodies 2D3 and 4F9 do not bind protein G and were purified by size-exclusion chromatography, the IgG antibody A6.2 was purified using a protein-A Sepharose column (GE Healthcare) according to the manufacturer's recommendations. All antibodies were dialyzed in PBS. Purity was assessed by SDS-PAGE and Coomassie staining, and protein concentration was determined using the BCA protein assay kit (Pierce).

**Virus stocks**. MNV-1.CW3 (GV/MNV1/2002/USA) was cultivated in BV-2 cells and used up to passage 6. Recombinant MNV viruses were produced using an adapted RNA-based reverse-genetics system[79] with the respective cDNA plasmid pT7MNV3'RZ to generate MNV-1.CW1. Recombinant viruses were sequence-confirmed for the entire VP1 ORF and used up to passage 3 post-transfection. Site-directed mutagenesis was performed using an adapted QuickChange approach[80] using the primer set (L386F: for: 5′ CTCTTGACTTCGTGGATGGCAGGGTTC GTGCGGTCCCAAGATCC 3′ and rev: 5′ GCCATCCACGAAGTCAAGAGAGG CCGCAGCAGTGACGCTG 3′) and (V378: for: 5′ TCGCCAGCTTCACTGCTG CGGCCTCTCTTGACTTGGTGGATGGC 3′ and rev: 5′ AGCAGTGAAGCTG GCGAACACCCTGCCCTGGTAGGGGGCC 3′). Virions were precipitated from clear-centrifuged cell lysate with a 45% saturated solution of ammonium sulfate (final concentration) and dialyzed into PBS. Viral titers were determined by plaque assay.

**Virus quantification by plaque assay**. The plaque assay was adapted from reference[76]. Briefly, BV-2 cells were seeded into six-well plates at a density of 5 × 10$^5$ cells per well and incubated overnight at 37 °C with 5% CO$_2$ and 95% humidity. For attachment, monolayers were inoculated with 500 μl virus stock serially diluted in PBS and incubated (37 °C, 5% CO$_2$) for 1 h, gently swirling the plates. After attachment, the inoculum was removed and cells were incubated for 48 h (37 °C, 5% CO$_2$) covered with 2 ml overlay medium containing 0.6% Avicel RC-591 (IMCD Deutschland GmbH) in DMEM supplemented with 10% (v/v) FCS (C-C-Pro), 20 mM HEPES (Biozym), 4 mM L-glutamine (Biozym), 0.2 mM non-essential amino acids (Biozym), and 200 units/ml penicillin and streptomycin (Biozym). After 48 h, the overlay medium was gently aspirated, cells were washed once with phosphate-buffered saline pH 7.4 (PBS) to remove residual Avicel. Cells were stained using erythrosine B staining solution (PBS with 1 mg/ml (w/v) erythrosine B). After 15–20 min incubation at room temperature, the staining solution was removed, and plaques were counted. Plaque assays were generally performed in duplicate and plaque-forming units (PFU) per ml were determined.

**Plaque neutralization assay**. The plaque neutralization assay is based on the plaque assay described above with the following modifications. For neutralization, a fixed concentration of 1 × 10$^6$ PFU of MNV was used. The serial dilution was conducted in 1× PBS in the presence of indicated additives. Specifically, bile acids (glycochenodeoxycholate GCDCA or taurocholate TCA, Sigma–Aldrich) were

added to each dilution to a final concentration of 500 μM and purified antibodies (2D3, 4F9, or A6.2) were added to each dilution to a final concentration of 2 μg per ml. The mixtures (500 μl) were applied to the cell monolayer and the standard plaque assay protocol was followed.

**Thermal stability assay**. The thermal stability assay is based on the plaque assay described above. Briefly, 1 × 10$^7$ PFU per ml of MNV-1.CW3 were incubated at temperatures ranging between 0 and 56 °C for 2 or 6 h in the presence or absence of 500 μM bile acid (sodium glycochenodeoxycholate GCDCA, Sigma–Aldrich) and immediately titered by plaque assay.

**Statistics and reproducibility**. Differences in plaque-forming units (PFU) are presented as means ± standard deviation (SD) of duplicate samples from at least three independent experiments. Statistical analysis was performed using a two-tailed $t$-test with two degrees of freedom. All statistical analyses were carried out using the Prism software package (GraphPad Software, CA).

**Reporting summary**. Further information on research design is available in the Nature Research Reporting Summary linked to this article.

## Data availability

The NMR assignment used in this paper has been published[40] and is deposited with the Biological Magnetic Resonance Data Bank (BMRB, https://bmrb.io) with the accession code 50919. Source data are provided with the paper as an Excel spread sheet (Supplementary Data 1). NMR raw data for methyl TROSY spectra used for TITAN analysis (Figs. 4 and 5) are provided as supplementary datasets (Supplementary Data 2, Supplementary Data 3).

## Code availability

Matlab and python scripts used for data analysis are available from the authors upon request.

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

## Acknowledgements

This research was funded by the Deutsche Forschungsgemeinschaft (DFG) via grants Pe494/12-2 (T.P.) and TA1093-2 (S.T.) within the research unit FOR2327 (ViroCarb). T.P. thanks the State of Schleswig-Holstein for supplying the NMR infrastructure (European Funds for Regional Development, LPW-E/1.1.2/857). C.F. thanks the Studienstiftung des deutschen Volkes for a fellowship. J.D. and C.U. acknowledge funding from FOR2327 ViroCarb (UE 183/1–2). C.U. and L.T. acknowledge funding from the EU Horizon 2020 project VIRUSCAN 731868. The Leibniz Institute for Experimental Virology (HPI) is supported by the Free and Hanseatic City of Hamburg and the Federal Ministry of Health. T.J.S. acknowledges funding from the NIH, grant 1R01-AI141465. We thank Cristiane Wobus at the University of Michigan (Michigan, USA) for the generous gift of the hybridoma cells 2D3, 4F9, and A6.2 and Ian Goodfellow (Cambridge University, United Kingdom) for the cDNA clone "pT7MNV3'RZ" for the optimized RNA-based reverse-genetics system. We would like to thank Thomas Krey, Institute of Biochemistry, University of Lübeck, for giving us access to the SEC-MALS apparatus.

## Author contributions

R.C., T.M., C.U., S.T., and T.P. designed experiments. R.C., T.M., A.M., C.F., and L.T.W. performed the NMR experiments. R.C., T.M., A.M., C.F., L.T.W., C.A.W., and T.P. analyzed and interpreted the NMR data. C.A.W. implemented new binding models into TITAN for data analysis and T.M. applied these models to the corresponding NMR datasets. R.C., T.M., C.F., and L.T.W. expressed and purified the isotope-labeled P-domains. R.C., T.M., and A.M. wrote Matlab and python scripts for data analysis. R.C. established the IEX protocol for the analysis of dissociation rates of P-dimers and performed the SEC and SEC-MALS as well as the differential scanning fluorimetry experiments. S.T. designed the virological experiments and V.H., M.S.L., and J.K. performed the experiments. T.J.S. interpreted data against the background of cryo-EM studies performed in his laboratory. J.D., L.T., and C.U. performed mass spectrometry experiments and analyzed the corresponding data. R.C., T.M., S.T., and T.P. wrote the paper. All authors contributed to data interpretation and commented on the manuscript.

## Funding

## Competing interests

The authors declare no competing interests.
