## [Peer Review File · Communications Biology]

Reviewers' comments:

Reviewer #1 (Remarks to the Author):

Brief summary of the manuscript

In the submitted manuscript, the authors elegantly apply NMR spectroscopy, mass spectroscopy, size exclusion chromatography and analytical ion exchange chromatography to probe Murine norovirus (MNV) plasticity and conformational changes observed on the MNV P-domain upon binding of glycochenodeoxycholic acid (GCDCA). These studies decipher the P-domain monomer: dimer equilibrium and specific role of GCDCA binding on P-domain dimerization. TROSY HSQC and methyl TROSY NMR studies demonstrate chemical shift changes upon GCDCA binding and reveal a slow exchange. The importance of this study is to improve our understanding on MHV: bile acid binding that leads to MNV P-domain allosteric conformational changes and immune escape. A major highlight of this paper is integration of various biophysical techniques and high-quality NMR data to understand bile acid-mediated binding effects on MNV P-domain dimerization. However, the manuscript suffers from over interpretation of some of the results, limitation of size exclusion chromatography, and complicated binding model. In addition, improvements in labeling of figures are required to improve manuscript visibility and reader understanding. Many aspects require additional consideration or re-interpretations which are described in the details below for manuscript revision.

Major points

Labeling of figures or color schemes used for crystal structure figures are not adequate in many places. For example, Fig2a shows six different chromatograms with various protein concentrations in the presence and absence of GCDCA. However, no labeling of chromatograms with various protein concentrations is provided. In addition, the concentration of GCDCA used for the chromatography study should be added in the figure panel or figure legends. Similarly, the GCDCA concentration used in HSQC and TROSY studies should be added in Fig3a and Fig3b legends. The authors should use two different color schemes for Fig3c crystal structures or other structures to differentiate P-domain protomers. Supplementary Fig. S2, which shows the sequence alignment of MNV P-domains is not clear. The authors should use web programs like Clustal Omega to perform multiple sequence alignment and show residue conservation, strong similarity by appropriate color scheme or by using asterisk and semicolons. In addition, the authors should also comment on murine and human norovirus P-domains similarity. Overall, better use of fonts and improvement in figures labeling are required throughout the whole manuscript.

Fig1c, which demonstrates size exclusion chromatogram at various feed concentrations. However, no column molecular standard chromatogram was provided to compare the data or demonstrate reproducibility or precision of the method. The authors also plot their SEC-derived binding data with feed concentrations that does not reflect the apparent concentration in the sizing column that carries a dilution factor. Furthermore, the use of analytical SEC is constrained to aberrations in protein elution characteristics that can be eliminated/minimized through other methods such as SEC-MALS that does not rely on interactions of the protein/protein oligomer with the column matrix. The combination of the use of feed concentration and SEC (over another method such as SEC-MALS) introduces concerns regarding the validity of the kinetic measurements presented. Another method that could be employed to extrapolate the clear monomer-dimer observation is isothermal calorimetry (ITC) that could provide solution affinity/kinetic data that does not rely on a separation method.

The authors use isothermal stability assays in Fig 2b to demonstrate CW1 P-domain stability which increases with increasing GCDCA concentrations. However, thermal stability assays of MNV-1.CW3 virions as described in Fig2c do not show enough statistical significance for all temperature ranges for 2hr or 6hr experiments. In addition, their measurements for thermal stability are based on an indirect functional assay and do not measure the thermal stability of the dimer specifically.

Ultimately, the functional outcome of bile salt binding is the most important finding, but the assay raises questions of potential non-P-domain-mediated bile acid effects. For better understanding of thermal stability of the P-domain dimer, the authors should consider more direct methods of thermal stability including differential scanning fluorimetry (DSF) and perform CW1 P-domain

stability in various concentrations of GCDCA; the method also allows for widespread screening/evaluation of a wide range of additives and at different concentrations to provide an enhanced analysis. DSF provides quantitative melting point (T_m) data to understand CW1 P-domain stability and T_m relations in the understanding of the monomer: dimer exchange.

In Fig 3b, the authors should explain whether both monomer and dimer peaks were observed in the apo (-GCDCA) methyl TROSY spectra, and how they were distinguished. Given that the protein concentration here is 50 μ M and $K_d=13\mu$ M from TITAN analysis, one would assume the unbound dimer "D" instead of monomer "M" form should dominate and be fitted in Fig 4b. This is confusing.

On page 14, the authors state that "This allows to estimate the exchange rate constant k_{ex} for binding of GCDCA to MNV P-dimers placing it in a range between 30 and 60 s⁻¹." However, the transition from slow to fast exchange limit due to peak separations requires order of magnitude differences. In fact, this is evidence for two different conformation exchange mechanism perhaps from different parts of the P-domain in HN-TROSY experiments. What are the numbers or percentage of peaks observed for fast and slow exchange?

Fig4, which describes the binding model for GCDCA binding and MNV P-domain dimerization is based on the overinterpretation and excessive normalization of NMR results. There is no evidence to distinguish DL and LD populations, (are there two sets of peaks in the NMR spectra of for apo-dimer?). The authors should simplify the equilibrium scheme and avoid differentiating DL and LD bound dimers with current data. It may be better denoted as DmL for being a mixed dimer with one apo and one ligand bound states. More importantly, the fitting of Fig 4b is poor or can even be better fitted by one straight line intersecting with the saturation limit. Could this suggest that the dominant kinetic rate is the interconversion to CD300If receptor-binding state, which unlocks the GCDCA binding sites?

Although some alanine methyl assignments are listed in Table S2, the authors should provide NMR assignments for all methyl groups shown in Fig. S5 and/or deposit the data in respective depositories such as the BMRB database?

Minor points

Page 4, the first three lines describe structural comparison between two crystal structures of MNV P-dimers. What was the resolution of these two crystal structures and what does RMSD of 0.495 Å exactly indicate to? Is it a C α -atoms similarity between all residues or does it refer to similarity of all atoms? The authors should use "PDB code" or "PDB ID" term in the manuscript to refer to mentioned crystal structures.

Supplementary Fig S1, the structural superimposition of open and close form would be more helpful specially to understand movement of E'F' and A'B' loops. What is the distance between these loops in open and close forms?

In Fig S3, the pH dependence of P-domain protein stability is clearly shown. Have the authors observed the same pH dependence for GCDCA bound P-domain dimer? On a related note, does the NMR spectrum revert to ap-form after successive addition and sequestration of GCDCA?

Fig S6 shows the spectra of various monomer-dimer signal pairs. What are the locations of those residues in the crystal structure? The authors should have an additional supplemental structural figure with locations of all amino acids analyzed in the manuscript. Are all these residues closer to the binding site of GCDCA? Is there any distance dependent shift between exchanging monomers and corresponding dimer cross peaks and location of these residues compared to GCDCA binding site?

Fig. S10, it would be better to plot the two sets of spectra for slow and fast exchanging residues at all four 0, 30, 75, and 300 μ M GCDCA concentrations for consistency in both panels a and b.

Reviewer #2 (Remarks to the Author):

Creutzmacher et al. describes protein NMR, native mass spec and analytical size exclusion chromatography studies conducted to explore the effects of bile acid GCDCA on MNV P-domain plasticity. The authors report monomer-dimer interconversion of MNV P-domains, with increasing protein concentration and presence of GCDCA associated with increased dimerization. GCDCA, but not other tested bile acids, is also associated with enhanced thermostability (of ~ 1 log PFU/mL) at 50 and 56 degrees. Chemical shift perturbation classifications indicate conformational changes both near the GCDCA binding pocket as well as remote allosteric effects in the E'F' loop upon GCDCA binding. This binding exclusively occurs to the dimer and not the monomer, and the authors derive a model wherein GCDCA binds the dimer in two independent binding events. Importantly, allosteric effects of GCDCA binding limit neutralizing antibody activity (directed towards the E'F' loop) against MNV. P-domain dimer dissociation rate constants are compared between MNV and a human NoV GII.4 strain; HNoV P-dimers are much more stable and have a much smaller koff. Overall, the data provided is compelling, the figures are well-arranged and clearly described in the well-written text, and the detailed methodological descriptions are likely to be useful to the field. This work adds nicely to a recently-growing body of literature exploring how bile acid regulates interaction of the MNV P-domain with the CD300lf receptor, explicitly addressing how this bile acid regulates dimerization and sensitivity to neutralizing antibodies that bind P2. Minor points should be addressed to add some additional clarification.

1. While the data in Fig 6D are compelling, did the authors also test the effects of GCDCA/TCA on A6.2-mediated neutralization? Presumably should show similar results?
2. Could the authors please demonstrate the locations of the antibody escape mutations discussed on page 4 (D348E, etc), perhaps as part of Fig S1? Illustrating this point visually would be helpful.
3. Additional feature labeling in Fig S2 would also be useful, such as P1 vs P2, surface loops, etc.
4. Gray and black curves should be more clearly labeled in Fig S8.
5. While a minor point, it seems that Fig 3 and 6 may make more sense to be grouped together in the manuscript, and perhaps Figs 4 and 5 could follow.
6. It would be useful to briefly discuss the in vivo infection implications of the bile acid/neutralizing antibody/P-domain plasticity interactions in the discussion.

Reviewer #3 (Remarks to the Author):

Creutzmacher et al. report their findings on the effect of GCDCA on the reversible equilibrium of MNV-P domains and associated conformational change that leads to antibody escape. To characterize the dynamic equilibrium and monitor the conformational change, the authors employed multiple approaches, including NMR, ITC, SEC, and native-MS.

Major findings are (1) MNV-P domains are in equilibrium between monomer and dimer, (2) dissociation rate constant of MNV-P dimer is faster than that of HuNoV protein GII.4 Saga, and (3) GCDCA-binding stabilize the dimer. Based upon the findings, the authors provide a model of immune escape upon binding of GCDCA. However, major shortcomings are noticed as follows.

1. First, this reviewer is not convinced that these findings are sufficiently new or interesting to warrant publication at Comms. Bio. The major criticism is that the information from most findings is rather redundant considering previous studies. For example, conformational change upon binding of GCDCA is already reported, and the monomer-dimer equilibrium of the P-domain is a basic property of any dimers. The mechanistic understanding of these phenomena is important. However, the findings described in the manuscript do not provide mechanisms underlying the dynamic equilibrium.
2. More fundamentally, it seems that the authors view the effect of GCDCA on the P-domain as an inducer or promoter; they use these terms repeatedly. Distinguishing the inducing and stabilizing

effect of a ligand is critical. Based on their own data, GCDCA stabilizes the dimer state. The authors did not provide any data supporting that GCDCA induces or promotes dimerization if it ever occurs.

3. On a similar note, the authors state (in Conclusion), "A picture emerges in which a small mediate molecule, a bile acid, triggers complex transformations in MNV but not in HuNoV capsids." Did they ever provide supporting evidence for this statement? Moreover, it is inappropriate to use "trigger." Again, the effect of the ligand should be stabilization of dimer; the authors did not provide any evidence of triggering effect of the ligand.

4. In addition, the authors state that the main focus of the manuscript is the "role of bile acids on MNV-P domain plasticity" (page 5). Unfortunately, this reviewer failed to find any data supporting the statement, i.e., none of the provided data is about it. To study the role of bile acids on domain plasticity, the authors have to conduct a comparative analysis of the conformational dynamics using order parameter analysis and/or micro-milliseconds dynamics between apo- and ligand-bound states. However, this manuscript only describes conformational change upon binding of GCDCA, which are mostly redundant from previous structural studies.

5. Although NMR is the most powerful method for characterizing dynamic equilibrium systems, the data presented in this manuscript falls short of rigor for warrant publication because of the following reasons.

6. A large portion of NMR analyses in this manuscript were conducted without rigorous validation of resonance assignment. Most of all, backbone resonances are not assigned at all. Resonances of monomer and dimer are under a mostly slow-exchange regime, according to the manuscript. In such a condition, using unassigned resonances is circumstantial at best and not suitable for any quantitative analysis. The authors should have limited their analysis only to the assigned resonances if any resonance was assigned. The authors previously did an assignment of human P-domains, indicating that it is feasible. It is not clear why the authors have not done it for MNV-P domains.

7. On a related topic, given the SEC and native-MS data, backbone-based data does not provide any further information on the equilibrium. The authors stretched the use of unassigned backbone data too much for mostly redundant information.

8. Sidechain methyl resonances were partly assigned by an indirect method using back-predicted noesy data based on a structure. The authors should provide a detailed description of the criteria used for selecting the 'correct' assignment. They also need to provide the protocol for validation of assignment.

9. The authors should provide the full assignment table in the supplementary information. Since the authors transferred the assignment to the apo-form, the assignment of the apo-form should be provided as well.

10. Conformational change of P-domain upon binding of GCDCA was monitored by CSP analysis. Although this analysis is typically powerful to follow a conformational change, its validity is uncertain without a proper resonance assignment of both apo- and ligand-bound forms. Especially, as the authors described, the resonance transfer of slow-exchanging resonances is not trivial. A very detailed protocol and validation procedure supporting their assignment transfer should be provided.

11. Moreover, their CSP analysis (Fig 3) could be compared with an RMSD analysis of two structures, i.e., apo- and ligand-bound forms. Both analyses present the conformational changes upon binding of GCDCA. Are they consistent? What additional information does CSP analysis provide?

12. In addition, the manuscript has a serious issue with references. Many references are neither complete nor inappropriate. For example, ref-40 is the pre-print version of the current manuscript. Surprisingly, the authors cite it to support their findings in the current manuscript. Moreover, Refs-9 and 19 are incomplete. The authors should carefully check the references.

Dear Editor, Dear Reviewers,

We thank the reviewers for carefully studying our manuscript and highlighting weaknesses. We have addressed all issues raised, and we believe that this has considerably improved the manuscript. At the time of submitting the first version of the manuscript we intended to present our data in concert with a cryo-EM study (Williams, A.N., Sherman, M.B., Smith, H.Q., Taube, S., Pettitt, B.M., Wobus, C.E., and Smith, T.J. (2021). A Norovirus Uses Bile Salts To Escape Antibody Recognition While Enhancing Receptor Binding. *J Virol* 95, e0017621) from Tom Smith's lab who is also a co-author on the present manuscript. The idea was to complement the solid-state study with solution-state data. The focus was then on the effect of the bile acid, GCDCA, on recognition of the murine norovirus capsid protein by neutralizing antibodies.

In the meantime, we have performed additional experiments as requested in part by the reviewers. The experiments include the assignment of ^{13}C -methyl resonance signals of so-called MILVA-labeled MNV P-domain, which has now been published (Maass, T., Westermann, L.T., Creutzmacher, R., Mallagaray, A., Dülfer, J., Uetrecht, C., and Peters, T. (2022). Assignment of Ala, Ile, Leu^{proS}, Met, and Val^{proS} methyl groups of the protruding domain of murine norovirus capsid protein VP1 using methyl–methyl NOEs, site directed mutagenesis, and pseudocontact shifts. Biomolecular NMR assignments). This additional step had forced us to scrutinize the provisional assignment presented with the first version of the manuscript. We realized that there were issues that were overlooked initially, and we would explicitly like to thank the reviewers for forcing us into this detailed analysis. It turned out that NMR analysis of MNV P-domains not only was impeded by the monomer-dimer equilibrium, but in addition the protein comes as a mixture of configurational isomers of Pro361 leading to extra resonance signals that we could not explain before. In the revised manuscript the NMR assignment is the basis for a much more detailed analysis of MNV P-dimer dissociation based on global line shape analysis with an algorithm developed by Chris Waudby from the University College in London. Given the complexity of the equilibria between P-domain monomers, dimers, and dimers with GCDCA bound, we decided to contact Chris and extend the binding models used for spectra simulation and line shape fitting. Two new binding models were implemented, one also considering potential cooperativity of binding. Improved data processing of methyl TROSY data sets allowed inclusion of more cross peaks into the fitting process for the analysis of dimer dissociation. We acknowledge that the error bars on the dissociation rate constant as presented in our original manuscript were not acceptable. We are very happy to have been given the opportunity to correct these shortcomings in our revised manuscript.

From this it becomes clear that the NMR part of the study has grown substantially, shifting the focus towards the biophysical characterization of the P-domain monomer-dimer equilibrium and especially towards the contrast between murine and human norovirus. This shift of focus and the inclusion of new data is reflected by an altered order of author's names and by adding Chris Waudby and two PhD students, one from Stefan Taube's lab, Miranda Sophie Lane, and another one from Charlotte Uetrecht's lab, Lars Thiede, to the list of authors. We hope that this is acceptable and that we can convince the reviewers and the editor that the revised manuscript is of sufficient quality to warrant publication in *Communications Biology*.

Thomas Peters on behalf of the author team

For a better understanding we have compiled the major changes applied to the manuscript before commenting and answering the reviewer's requests:

New experiments and extended data analysis in the revised manuscript:

1. We have added native MS data for MNV CW1 P-domains, replacing the data for MNV CR10 P-domains in Fig. 1d to match the NMR data in the same figure, which have also been acquired with MNV CW1. The native MS spectra for CR10 are now found in Fig. S6.
2. SEC-MALS experiments (Fig. S5) have been performed.
3. Differential Scanning Fluorimetry (DSF) measurements to better define thermal stability of P-dimers (Fig. 3b and Fig. S11).
4. Assignment of ^{13}C -methyl groups of MILVA-labeled MNV CW1 P-dimers saturated with GCDCA (Maass, T., Westermann, L.T., Creutzmacher, R., Mallagaray, A., Dülfer, J., Uetrecht, C., and Peters, T. (2022). Assignment of Ala, Ile, Leu^{proS}, Met, and Val^{proS} methyl groups of the protruding domain of murine norovirus capsid protein VP1 using methyl–methyl NOEs, site directed mutagenesis, and pseudocontact shifts. Biomolecular NMR assignments) and a more exhaustive transfer of the assignments to the apo-form of the P-domain (Table S3).
5. The concentration dependent methyl TROSY spectra (Fig. S6 old manuscript, Fig. S9 revised manuscript) have been re-processed with NMRPipe using baseline corrections in both dimensions. During re-processing we detected a wrong setting for the factor scaling the coupling constant for spectra simulation with TITAN. The missing baseline correction and the wrong setting for the delay $1/2J$ had caused a large standard deviation in the initial analysis. This has now been corrected. For the new global line shape analysis with TITAN we included 13 instead of 7 cross peaks, yielding much better standard deviations for the derived dissociation constant and the dissociation rate constant (see Table S2 revised manuscript).
6. We have performed a global line shape analysis with TITAN for the titration of MNV CW1 P-domain with GCDCA. This analysis (Table S2 of the revised manuscript) substitutes the former analysis based on intensities of ^{13}C methyl cross peaks in slow exchange (Fig. S12 and Fig. 4 of the old manuscript). For the new analysis, two binding models were implemented with TITAN, one model consisting of a simple two-step binding process and the other one considering possible cooperativity (Fig. 5 and Fig. S20 of the revised manuscript). The results of the new global line shape analysis are compiled in Table S2.

Experiments and (redundant) data removed in the revised manuscript:

1. Quantitative data analysis based on non-assigned backbone ^1H , ^{15}N TROSY HSQC spectra has been removed.
2. The analysis of GCDCA binding to MNV P-dimers based on the intensities of slowly exchanging cross peaks has been substituted by a new global line shape analysis (see above 4.).
3. The quantitative analysis of tailing peaks of SEC runs to derive dissociation constants has been removed (Eqs. 2-8 of the old manuscript).
4. The analysis of TRACT experiments has been removed as it provides redundant information (Fig. S8 old manuscript).

In the following we respond to each of the points raised:

Requests by the editor:

NMR data analysis and in particular resonance assignment should be revisited in light of reviewer 3's comments.

We have taken care of this point by finishing the assignment of MILVA-labeled NMR P-domain and publishing it in the journal "Biomolecular NMR Assignments" prior to revising the manuscript. See also the more detailed explanations given above.

Overstatements and overinterpretation of the results should be removed.

We have avoided overinterpretations and overstatements. In this respect, the shift of focus (see above) has been supportive.

Additional biophysical studies of the monomer-dimer equilibrium as suggested by reviewers 1 and 3 should be included if possible.

We have done almost all the experiments requested (for details see below).

All minor issues raised by the reviewers should be suitably addressed.

We have tried our best.

Reviewer #1 (Remarks to the Author):

Brief summary of the manuscript

In the submitted manuscript, the authors elegantly apply NMR spectroscopy, mass spectroscopy, size exclusion chromatography and analytical ion exchange chromatography to probe Murine norovirus (MNV) plasticity and conformational changes observed on the MNV P-domain upon binding of glycochenodeoxycholic acid (GCDCA). These studies decipher the P-domain monomer: dimer equilibrium and specific role of GCDCA binding on P-domain dimerization. TROSY HSQC and methyl TROSY NMR studies demonstrate chemical shift changes upon GCDCA binding and reveal a slow exchange. The importance of this study is to improve our understanding on MHV: bile acid binding that leads to MNV P-domain allosteric conformational changes and immune escape. A major highlight of this paper is integration of various biophysical techniques and high-quality NMR data to understand bile acid-mediated binding effects on MNV P-domain dimerization. However, the manuscript suffers from over interpretation of some of the results, limitation of size exclusion chromatography, and complicated binding model. In addition, improvements in labeling of figures are required to improve manuscript visibility and reader understanding. Many aspects require additional consideration or re-interpretations which are described in the details below for manuscript revision.

Major points

Labeling of figures or color schemes used for crystal structure figures are not adequate in many places. For example, Fig2a shows six different chromatograms with various protein concentrations in the presence and absence of GCDCA. However, no labeling of chromatograms with various protein concentrations is provided. In addition, the concentration of GCDCA used for the chromatography study should be added in the figure panel or figure legends.

We have added the labeling and the GCDCA concentrations are included. Fig. 2a has become Fig. 3a now. In fact, we have revised all figures containing crystal structures in both the main text and the supplementary information

Similarly, the GCDCA concentration used in HSQC and TROSY studies should be added in Fig3a and Fig3b legends.

Fig. 3 has become Fig. 4 now. The ^1H , ^{15}N TROSY HSQC spectrum has been removed since it does not add new information at this point. The methyl TROSY spectrum is shown in Fig. 4a (has been Fig. 3b before) and the protein concentrations are given in the legend. Compared to the original Fig. 3, the information presented here is more concise, and easier to understand (in our opinion).

The authors should use two different color schemes for Fig3c crystal structures or other structures to differentiate P-domain protomers.

As said above, this figure is completely "refurbished". We think that the new cartoon allows easier discrimination between the monomers. We have refrained from using different color schemes since this may be misinterpreted as distinct CSPs.

Supplementary Fig. S2, which shows the sequence alignment of MNV P-domains is not clear. The authors should use web programs like Clustal Omega to perform multiple sequence alignment and show residue conservation, strong similarity by appropriate color scheme or by using asterisk and semicolons.

We had already used Clustal Omega for preparing Fig. S2, but we agree that the graphical representation was not well readable. We improved this in the revised version of the manuscript.

In addition, the authors should also comment on murine and human norovirus P-domains similarity.

This is an important point. A structure alignment is presented in Fig. S10 reflecting the high overall structural similarity. We have also subjected human and murine norovirus P-dimers to an interaction analysis using the PRODIGY server. The results are shown in Table S6.

Overall, better use of fonts and improvement in figures labeling are required throughout the whole manuscript.

We have done our best to achieve this goal.

Fig1c, which demonstrates size exclusion chromatogram at various feed concentrations. However, no column molecular standard chromatogram was provided to compare the data or demonstrate reproducibility or precision of the method.

We apologize for not showing the calibration used. The calibration is now shown in Fig. S4, and experimental details are included with the methods section. We decided to use the SEC data only for an estimate of the range of dissociation constants and, therefore, moved the binding isotherm (formerly part of Fig. 1c) to the supplementary material (Fig. S6). In Fig. 1 we also acquired a new native mass spectrum for MNV CW1 to make the data more consistent with the NMR part. The mass spectrum for MNV CR10 is now found in the supplementary material as part of Fig. S6.

The authors also plot their SEC-derived binding data with feed concentrations that does not reflect the apparent concentration in the sizing column that carries a dilution factor.

We agree. The protein gets diluted from 10 μ l to roughly 100 μ l standard peak width. Therefore, and as pointed out above, we used the SEC data only for an estimate of the dissociation constant. Curve fitting using the two boundary conditions (feed volume and feed volume/10) yields dissociation constants of 4 and 40 μ M, respectively, which agrees with the results from the other biophysical methods applied here (see Table S2).

Furthermore, the use of analytical SEC is constrained to aberrations in protein elution characteristics that can be eliminated/minimized through other methods such as SEC-MALS that does not rely on interactions of the protein/protein oligomer with the column matrix. The combination of the use of feed concentration and SEC (over another method such as SEC-MALS) introduces concerns regarding the validity of the kinetic measurements presented.

We have performed SEC-MALS experiments, supporting our initial conclusions based on elution behavior. The SEC-MALS data are shown in Fig. S5. However, we removed the kinetic analysis based on peak-tailing. We agree with the reviewer that this analysis is based on too many assumptions. In the light of new and much better reproducible kinetic data from NMR titration analysis this becomes obsolete anyway.

Another method that could be employed to extrapolate the clear monomer-dimer observation is isothermal calorimetry (ITC) that could provide solution affinity/kinetic data that does not rely on a separation method.

Kinetic ITC is not a "direct" method but relies on fitting kinetic models to experimentally measured heats. In contrast, we have direct evidence from two independent methods, NMR, and native MS, clearly showing the presence of monomers and dimers in an equilibrium.

The authors use isothermal stability assays in Fig 2b to demonstrate CW1 P-domain stability which increases with increasing GCDCA concentrations. However, thermal stability assays of MNV-1.CW3 virions as described in Fig2c do not show enough statistical significance for all temperature ranges for 2hr or 6hr experiments. ...

Temperatures in the physiological range and below do not affect MNV-1.CW3 infectivity, we therefore do not expect an effect of GCDCA in this temperature range. However, temperature above 45 $^{\circ}$ C affects infectivity, and at these elevated temperatures a protective effect of GCDCA is observed. With this

observation we would like to link the results of the *in vitro* thermal stability assays (DSF, HIC) to infectivity. Longer incubation times above 56 °C are detrimental to the virions and measurements are not possible. We think it is important to retain these data as thermal stability on a molecular level correlates with thermal stability of the virions as reflected by infectivity. The graph is now placed with the supplementary material (Fig. S11).

... In addition, their measurements for thermal stability are based on an indirect functional assay and do not measure the thermal stability of the dimer specifically.

The HIC based data are also moved to the supplementary part (Fig. S11) and are substituted by direct stability measurements using differential scanning fluorimetry (DSF) experiments, now shown in Fig. 3b. It should be noted that the results from "direct" DSF and from "indirect" HIC experiments are rather similar (Fig. S11).

Ultimately, the functional outcome of bile salt binding is the most important finding, but the assay raises questions of potential non-P-domain-mediated bile acid effects

As alluded to above we have added substantially more experimental evidence to characterize GCDCA-mediated stabilization of MNV P-dimers. In the light of our new data and the published cryo-EM study there is no doubt that it is the interaction of the bile acid with the P-dimer (not the monomer!) that is responsible for P-dimer stabilization.

For better understanding of thermal stability of the P-domain dimer, the authors should consider more direct methods of thermal stability including differential scanning fluorimetry (DSF) and perform CW1 P-domain stability in various concentrations of GCDCA; the method also allows for widespread screening/evaluation of a wide range of additives and at different concentrations to provide an enhanced analysis. DSF provides quantitative melting point (T_m) data to understand CW1 P-domain stability and T_m relations in the understanding of the monomer: dimer exchange.

We agree and have performed DSF experiments (Fig. 3b and Fig. S11). See comments above.

In Fig 3b, the authors should explain whether both monomer and dimer peaks were observed in the apo (-GCDCA) methyl TROSY spectra, and how they were distinguished. Given that the protein concentration here is 50 μ M and $K_d=13\mu$ M from TITAN analysis, one would assume the unbound dimer "D" instead of monomer "M" form should dominate and be fitted in Fig 4b. This is confusing.

We have completely removed this part describing the calculation of a dissociation constant from signal intensities via numerically solving eqs. 14 and 15 (old manuscript). In the revised manuscript analysis of the coupled equilibria is solely based on TITAN analysis (Fig. S9, Table S2). In the previous version of the manuscript methyl TROSY spectra had been processed without applying baseline corrections and spectra simulations employed a wrong setting of the delay during which antiphase magnetization develops and is converted into multiple quantum coherence (see also initial general remarks). We have corrected this, yielding a similar value for dissociation constant $K_{D,Dimer}$ (7 μ M in Table S2 of the revised manuscript vs. 12 μ M in Table 1 of the old manuscript). However, as mentioned already above the off-rate constant k_{off} in the old manuscript had a rather large error bar of the order of the value itself (legend to Fig. S6 of the old manuscript), rendering it unreliable. Therefore, the old value reported for k_{off} must be considered as invalid. We apologize for having overlooked this obvious fault. The corrected analysis yields acceptable standard deviations as seen from Table S2 in the revised manuscript.

On page 14, the authors state that "This allows to estimate the exchange rate constant k_{ex} for binding of GCDCA to MNV P-dimers placing it in a range between 30 and 60 s⁻¹." However, the transition from slow to fast exchange limit due to peak separations requires order of magnitude differences. In fact, this is evidence for two different conformation exchange mechanism perhaps from different parts of the P-domain in HN-TROSY experiments. What are the numbers or percentage of peaks observed for fast and slow exchange?

This statement is not correct, indeed. We incorrectly labeled the cross peak of the HN-TROSY in Fig. S10b as "fast exchange", which it is not. This is corrected in the revised manuscript, now Fig. S19, where it is correctly labeled as "intermediate exchange". We agree with the reviewer that estimation of dissociation rate constants based on these data is not convincing and removed this statement like we have and removed almost all discussion based on backbone NMR data. We have left Fig. S19 in the supplementary information just for illustration. In the revised manuscript we base our arguments on a new TITAN analysis of methyl TROSY titration data, leading to a dissociation rate constant for dissociation of GCDCA from MNV P-dimers of 28 Hz (Table S2).

Fig4, which describes the binding model for GCDCA binding and MNV P-domain dimerization is based on the overinterpretation and excessive normalization of NMR results.

As already mentioned, the analysis based on the binding model shown in Fig. 4 of the original manuscript has been removed and is substituted by the new TITAN analysis as visualized in Fig. 5 and Fig. S9.

There is no evidence to distinguish DL and LD populations, (are there two sets of peaks in the NMR spectra of apo-dimer?). The authors should simplify the equilibrium scheme and avoid differentiating DL and LD bound dimers with current data. It may be better denoted as DmL for being a mixed dimer with one apo and one ligand bound states.

We have completely redone this analysis as summarized in Fig. 5. We have avoided the use of DL/LD as, indeed, we cannot discriminate between the two forms. We have tested whether there is indirect evidence of such states by applying a cooperative binding model with the TITAN analysis. However, no indication for such a mechanism can be found. Therefore, the simple binding model is best describing our observations.

More importantly, the fitting of Fig 4b is poor or can even be better fitted by one straight line intersecting with the saturation limit. Could this suggest that the dominant kinetic rate is the interconversion to CD300If receptor-binding state, which unlocks the GCDCA binding sites?

As mentioned, we have completely removed this part.

Although some alanine methyl assignments are listed in Table S2, the authors should provide NMR assignments for all methyl groups shown in Fig. S5 and/or deposit the data in respective depositories such as the BMRB database?

The assignment is now published, and the data is deposited with the BioMagResBank (see manuscript for details).

Minor points

Page 4, the first three lines describe structural comparison between two crystal structures of MNV P-dimers. What was the resolution of these two crystal structures and what does RMSD of 0.495 Å exactly indicate to? Is it a C α -atoms similarity between all residues or does it refer to similarity of all atoms? The authors should use "PDB code" or "PDB ID" term in the manuscript to refer to mentioned crystal structures.

The alignment of the C α atoms of the P-domains of has been done with PyMOL using the following sequence of commands:

```
fetch 6c6q 6e47, async=0
align 6e47 & c. A & n. ca, 6c6q & c. B & n. ca, object=Pdimer_calpha
```

We thank the reviewer for this remark as in the original calculus it were not the C α atoms that were used for alignment (although they should have been used). The correct value of 0.132 Å for the C α RMSD is now reported in the manuscript. The resolution of the crystal structures is as follows: 1.95 Å for 6e47, and 2.00 Å for 6c6q. However, there is a general issue with RMSD values derived from different

alignment programs as they use different alignment algorithms in the first step of sequence alignment. PyMOL uses CLUSTAL and ChimeraX has the Needleman-Wunsch algorithm as a default. Therefore, we included a reference to PyMOL and included the lines of code in the methods section. This guarantees that everyone can reproduce the values and go into the details of how they were obtained, if desired. We have now only used capital letters PDB.

Supplementary Fig S1, the structural superimposition of open and close form would be more helpful specially to understand movement of E'F' and A'B' loops. What is the distance between these loops in open and close forms?

We have now both presentations, side-by-side and superimposed, included with Fig. S1. The distance between the loops in the open and closed form has been added to the legend, by using the distances between individual C α atoms in the loops as examples.

In Fig S3, the pH dependence of P-domain protein stability is clearly shown. Have the authors observed the same pH dependence for GCDCA bound P-domain dimer? On a related note, does the NMR spectrum revert to ap-form after successive addition and sequestration of GCDCA?

The pH-dependence of the stability of MNV P-dimers with GCDCA bound has not been tested. However, we have almost completed a comprehensive study on the influence of pH and bivalent metal ion binding on the stability and dissociation kinetics of MNV P-dimers. This is follow-up work that will be published separately. Briefly, pH and metal-ion binding have a significant influence on the stability of apo P-dimers but much less on GCDCA-bound P-dimers. We have included Fig. S14, showing that the apo-state is fully recovered after removing GCDCA via SEC chromatography.

Fig S6 shows the spectra of various monomer-dimer signal pairs. What are the locations of those residues in the crystal structure? The authors should have an additional supplemental structural figure with locations of all amino acids analyzed in the manuscript. Are all these residues closer to the binding site of GCDCA? Is there any distance dependent shift between exchanging monomers and corresponding dimer cross peaks and location of these residues compared to GCDCA binding site?

The amino acids used for line-shape analysis are now mapped on the P-dimer in Fig. S9 (formerly Fig. S6). Four other amino acids that have been used for line-shape analysis have not been assigned. The concentration-dependent spectra used for Fig. S9 (formerly S6) contain only apo P-domains. To be honest, I don't understand the meaning of the last sentence.

Fig. S10, it would be better to plot the two sets of spectra for slow and fast exchanging residues at all four 0, 30, 75, and 300uM GCDCA concentrations for consistency in both panels a and b.

We have complemented this figure (now Fig. S19) with a corresponding figure showing methyl TROSY with varying concentrations of GCDCA (Fig. S8). We would like to leave S8 as it is. It is only meant to visualize the different regimes of chemical exchange. The picture simply becomes clearer when using the concentrations chosen. It should be noted that the exchange shown in Fig. S19b is not "fast" as stated in the original manuscript. The intensity of the peak changes along the titration, clearly indicating intermediate exchange or at best fast-to-intermediate exchange. We have corrected this in the revised manuscript. N.B.: There are hardly any peaks in fast exchange, neither in the backbone nor in the methyl TROSY spectra, which is fully consistent with a dissociation rate constant of 26 Hz for the dissociation of GCDCA (Table S2).

Reviewer #2 (Remarks to the Author):

Creutzmacher et al. describes protein NMR, native mass spec and analytical size exclusion chromatography studies conducted to explore the effects of bile acid GCDCA on MNV P-domain plasticity. The authors report monomer-dimer interconversion of MNV P-domains, with increasing protein concentration and presence of GCDCA associated with increased dimerization. GCDCA, but not other tested bile acids, is also associated with enhanced thermostability (of ~1 log PFU/mL) at 50 and 56 degrees. Chemical shift perturbation classifications indicate conformational changes both near the GCDCA binding pocket as well as remote allosteric effects in the E'F' loop upon GCDCA binding. This binding exclusively occurs to the dimer and not the monomer, and the authors derive a model wherein GCDCA binds the dimer in two independent binding events. Importantly, allosteric effects of GCDCA binding limit neutralizing antibody activity (directed towards the E'F' loop) against

MNV. P-domain dimer dissociation rate constants are compared between MNV and a human NoV GII.4 strain; HNoV P-dimers are much more stable and have a much smaller koff. Overall, the data provided is compelling, the figures are well-arranged and clearly described in the well-written text, and the detailed methodological descriptions are likely to be useful to the field. This work adds nicely to a recently-growing body of literature exploring how bile acid regulates interaction of the MNV P-domain with the CD300lf receptor, explicitly addressing how this bile acid regulates dimerization and sensitivity to neutralizing antibodies that bind P2. Minor points should be addressed to add some additional clarification.

1. While the data in Fig 6D are compelling, did the authors also test the effects of GCDCA/TCA on A6.2-mediated neutralization? Presumably should show similar results?

We have included data on A6.2 mediated neutralization. The results are in accordance with the data published recently by Williams et al., 2021. As mentioned above, the original plan had been to have our solution state data published in concert with the cryo-EM study. Our data support the published data showing that both, GCDCA and TCA decrease the affinity of Fabs 2D3, 4F9, and A6.2 for binding to the P-domain, but with TCA being much less efficacious than GCDCA (Fig. S18).

2. Could the authors please demonstrate the locations of the antibody escape mutations discussed on page 4 (D348E, etc), perhaps as part of Fig S1? Illustrating this point visually would be helpful.

This has been done in Fig. S1.

3. Additional feature labeling in Fig S2 would also be useful, such as P1 vs P2, surface loops, etc.

This figure has been improved according to the request of reviewer #1. The requested labeling has been added in Fig. S16.

4. Gray and black curves should be more clearly labeled in Fig S8.

The TRACT experiments have been removed as they provide only redundant information.

5. While a minor point, it seems that Fig 3 and 6 may make more sense to be grouped together in the manuscript, and perhaps Figs 4 and 5 could follow.

These figures have been completely revised and are more concise now. Fig. 3 in the old manuscript is Fig. 4 in the revised manuscript. Fig. 6 of the old manuscript has been removed because of shifting the focus. Fig. 4 old is now Fig. 5 revised. Fig. 5 old has become Fig. 2 revised.

6. It would be useful to briefly discuss the in vivo infection implications of the bile acid/neutralizing antibody/P-domain plasticity interactions in the discussion.

As immune escape is no longer the focus, we have discussed the potential biological consequences of fast dimer dissociation of MNV P-dimers in the absence of GCDCA and stable MNV P-dimers when

GCDCA is bound in the revised manuscript. We think this adds a novel perspective not really sensed by the community before.

Reviewer #3 (Remarks to the Author):

Cruetzner et al. report their findings on the effect of GCDCA on the reversible equilibrium of MNV-P domains and associated conformational change that leads to antibody escape. To characterize the dynamic equilibrium and monitor the conformational change, the authors employed multiple approaches, including NMR, ITC, SEC, and native-MS. Major findings are (1) MNV-P domains are in equilibrium between monomer and dimer, (2) dissociation rate constant of MNV-P dimer is faster than that of HuNoV protein GIL4 Saga, and (3) GCDCA-binding stabilize the dimer. Based upon the findings, the authors provide a model of immune escape upon binding of GCDCA. However, major shortcomings are noticed as follows.

1. First, this reviewer is not convinced that these findings are sufficiently new or interesting to warrant publication at *Comms. Bio.* The major criticism is that the information from most findings is rather redundant considering previous studies. For example, conformational change upon binding of GCDCA is already reported, ...

Conformational changes of MNV P-domains upon binding of GCDCA have NOT been reported before. It had been observed before that the E'F' loop can be oriented in two positions (Taube et al., 2010). However, this was independent of GCDCA, which was absent in that study. At the time, it was only a hypothesis that GCDCA may induce a change in the E'F' loop orientation leading to immune escape. The first description that binding of GCDCA is causing immune escape because of the E'F' loop reorienting has been based on cryo-EM data and is now published (Williams et al., 2021). In fact, we had planned to have the NMR study published along with the cryo-EM study as it complemented the cryo-EM data with solution state data, directly showing CSPs in the E'F' loop upon addition of GCDCA, in accordance with the loop reorientation.

We have now shifted the focus of the paper towards the description of the monomer-dimer equilibrium of P-dimers and the contrast between human and murine noroviruses, as pointed out above in the general remarks to the editor and the reviewers. We think that these findings will have an impact on the understanding of how norovirus capsid stability can be modulated.

... and the monomer-dimer equilibrium of the P-domain is a basic property of any dimers. The mechanistic understanding of these phenomena is important. However, the findings described in the manuscript do not provide mechanisms underlying the dynamic equilibrium.

The old manuscript had presented biophysical data reflecting fundamental differences in dimer formation and dissociation for human and murine norovirus P-dimers. Arguably, the data set allowed limited insight into the "mechanism" of dimer formation and the analysis was incoherent in parts. In the meantime, we have improved the analysis of the NMR experiments, and we performed additional biophysical experiments (see the corresponding answers to reviewer #1). We extended the analysis of dynamic NMR data into GCDCA binding, now allowing to put forward a proposal for a three-state binding model, where in a first step P-domain monomers must form dimers, before in a second step GCDCA can bind, leading to stable P-dimers (see Fig. 5 and the corresponding discussion in the revised manuscript). We have employed state-of-the-art NMR methods to analyze norovirus P-dimers as far as currently possible, arriving at a set of dissociation rate constants and dissociation constants for P-dimer dissociation and for binding of GCDCA. These new data are the basis for understanding any underlying mechanism.

2. More fundamentally, it seems that the authors view the effect of GCDCA on the P-domain as an inducer or promoter; they use these terms repeatedly. Distinguishing the inducing and stabilizing effect

of a ligand is critical. Based on their own data, GCDCA stabilizes the dimer state. The authors did not provide any data supporting that GCDCA induces or promotes dimerization if it ever occurs.

We have removed the terms "inducer" and "promoter" and agree that this may have too far-fetched mechanistic implications. While performing additional experiments and extending data analysis we made every effort to phrase the results and the corresponding discussion such that the wording matches the experimental facts as close as possible. We hope that the new text convinces the reviewer.

3. On a similar note, the authors state (in Conclusion), "A picture emerges in which a small mediate molecule, a bile acid, triggers complex transformations in MNV but not in HuNoV capsids." Did they ever provide supporting evidence for this statement?

The evidence is that there is no comparable binding pocket for GCDCA in human norovirus P-dimers. Therefore, GCDCA cannot stabilize human P-dimers in the way this is observed for murine norovirus P-dimers. In addition, we show that human norovirus P-dimers are way more stable than their murine counterparts in the absence of GCDCA. We have tried to make this much more transparent in the revised manuscript. The data can now also be viewed in the light of the structural details as revealed by the cryo-EM study of Williams et al., 2021.

Moreover, it is inappropriate to use "trigger." Again, the effect of the ligand should be stabilization of dimer; the authors did not provide any evidence of triggering effect of the ligand.

We removed any mention of "trigger" as it may suggest mechanistic details the NMR experiments cannot directly demonstrate. Still, the long-range NMR CSP effects observed may be interpreted as being associated with corresponding conformational transitions in the respective (loop) regions. Clearly, a CSP can never "prove" a conformational change as it may be due to very subtle changes in the chemical environment. However, in view of the cryo-EM data by Williams et al., 2021 we dare to say that binding of GCDCA not only stabilizes the dimers but also causes remote effects such as the loop reorientation, which can be correlated with the CSPs observed. In the revised manuscript we have tried to avoid any exaggerated speculations into this direction.

4. In addition, the authors state that the main focus of the manuscript is the "role of bile acids on MNV-P domain plasticity" (page 5). Unfortunately, this reviewer failed to find any data supporting the statement, i.e., none of the provided data is about it. To study the role of bile acids on domain plasticity, the authors have to conduct a comparative analysis of the conformational dynamics using order parameter analysis and/or micro-milliseconds dynamics between apo- and ligand-bound states. However, this manuscript only describes conformational change upon binding of GCDCA, which are mostly redundant from previous structural studies.

We have removed the term "plasticity" throughout the manuscript.

5. Although NMR is the most powerful method for characterizing dynamic equilibrium systems, the data presented in this manuscript falls short of rigor for warrant publication because of the following reasons.

6. A large portion of NMR analyses in this manuscript were conducted without rigorous validation of resonance assignment. Most of all, backbone resonances are not assigned at all. Resonances of monomer and dimer are under a mostly slow-exchange regime, according to the manuscript. In such a condition, using unassigned resonances is circumstantial at best and not suitable for any quantitative analysis. The authors should have limited their analysis only to the assigned resonances if any resonance was assigned. The authors previously did an assignment of human P-domains, indicating that it is feasible. It is not clear why the authors have not done it for MNV-P domains.

The revised manuscript now builds on a published assignment of ¹³C-methyl groups in a MILVA labeled sample of MNV P-domain (Maas et al., 2022). We have reduced analysis of backbone NMR data to a minimum as an assignment is not available. The assignment of the human norovirus P-domain was a

challenging project given the size of the protein and the issues of spontaneous post-translational modification that came with it (Creutzmacher et al., Nature Communications 2019). The murine counterpart is more than a challenge. We have not been able to develop a refolding protocol that can be used for synthesizing triple-labeled samples at reasonable costs (the protocol developed for human P-domains was useless for the current system). Therefore, we focused on ¹³C-methyl labeling, allowing us to measure dissociation rate constants and dissociation constants for the formation of dimers and for binding of GCDCA. As mentioned, we are very happy to have been offered the opportunity to finalize this assignment because it turned out that MNV P-domains carry a proline residue, Pro361, that comes as a mixture of cis- and trans-isomers. We admit, that in the first version of this manuscript we have not paid enough attention to a rigorous treatment and overlooked this effect. The conclusions arrived at in the revised manuscript are based on more rigorous data analysis and on extended NMR data sets. We also performed additional biophysical experiments as requested by reviewer #1.

7. On a related topic, given the SEC and native-MS data, backbone-based data does not provide any further information on the equilibrium. The authors stretched the use of unassigned backbone data too much for mostly redundant information.

We would like to ask reviewer #3 to look at the reply to reviewer #1 who had requested more biophysical experiments such as SEC-MALS and DSF. We have performed these experiments and included the data with the revised manuscript. As for the NMR data our arguments are now based on an extended set of data and on a more sophisticated NMR data analysis using global line shape analysis. Unassigned backbone data have no longer been used to come to quantitative conclusions.

8. Sidechain methyl resonances were partly assigned by an indirect method using back-predicted noesy data based on a structure. The authors should provide a detailed description of the criteria used for selecting the 'correct' assignment. They also need to provide the protocol for validation of assignment.

The assignment has just been published in "Biomolecular NMR Assignments" (Maass et al., 2022).

9. The authors should provide the full assignment table in the supplementary information. Since the authors transferred the assignment to the apo-form, the assignment of the apo-form should be provided as well.

In the revised manuscript we explain how partial assignments were transferred from the GCDCA bound form of the MNV P-domain to the apo form. The results are summarized in Table S3.

10. Conformational change of P-domain upon binding of GCDCA was monitored by CSP analysis. Although this analysis is typically powerful to follow a conformational change, its validity is uncertain without a proper resonance assignment of both apo- and ligand-bound forms. Especially, as the authors described, the resonance transfer of slow-exchanging resonances is not trivial. A very detailed protocol and validation procedure supporting their assignment transfer should be provided.

See reply to 9. above.

11. Moreover, their CSP analysis (Fig 3) could be compared with an RMSD analysis of two structures, i.e., apo- and ligand-bound forms. Both analyses present the conformational changes upon binding of GCDCA. Are they consistent? What additional information does CSP analysis provide?

We have performed a C α RMSD analysis of the crystal structures PDB 3lq6 (apo) and PDB 6e47(GCDCA-bound) in Fig. S16. It is seen that the largest RMSD values are found in the loop regions. This correlates well with long-range CSPs observed in methyl TROSY spectra (see manuscript, e.g., Fig. 3 and accompanying discussion). Additional CSPs in the C-terminal region may hint towards yet unidentified effects of adding GCDCA. We note that we observe CSPs in regions where crystal structure models of the apo and bound state do not show any differences, indicating that CSP analysis can uncover subtle effects in solution that are absent in solid-state crystal structure models.

12. In addition, the manuscript has a serious issue with references. Many references are neither complete nor inappropriate. For example, ref-40 is the pre-print version of the current manuscript. Surprisingly, the authors cite it to support their findings in the current manuscript. Moreover, Refs-9 and 19 are incomplete. The authors should carefully check the references.

We apologize for having been so inattentive writing the original version of the manuscript. All errors and inaccuracies are now corrected.

Reviewers' comments:

Reviewer #1 (Remarks to the Author):

I agree with the authors, revised manuscript has improved a lot and addition of native MS data, SEC-MALS data in Fig S5, reprocessing of TROSY spectra, improved structural figures and BMRB assignment provide a lot of clarity. Kudos to the authors for addressing all of the major concerns and additional considerations. Only minor suggestion I will have is the addition of Clustal Omega citation in Fig S2. I advise any of the relevant citation with the following :

Fast, scalable generation of high-quality protein multiple sequence alignments using Clustal Omega Sievers F, Wilm A, Dineen DG, Gibson TJ, Karplus K, Li W, Lopez R, McWilliam H, Remmert M, Söding J, Thompson JD, Higgins D Molecular Systems Biology 7 Article number: 539 doi:10.1038/msb.2011.75

A new bioinformatics analysis tools framework at EMBL-EBI (2010) Goujon M, McWilliam H, Li W, Valentin F, Squizzato S, Paern J, Lopez R Nucleic acids research 2010 Jul, 38 Suppl: W695-9 doi:10.1093/nar/gkq313

Analysis Tool Web Services from the EMBL-EBI. (2013) McWilliam H, Li W, Uludag M, Squizzato S, Park YM, Buso N, Cowley AP, Lopez R Nucleic acids research 2013 Jul;41(Web Server issue):W597-600 doi:10.1093/nar/gkt376

Reviewer #3 (Remarks to the Author):

Creutzmacher et al. describe their findings on the monomer-dimer equilibrium of the MNV-P domain and how a ligand affects the equilibrium. Although this revised manuscript is improved compared to the original version, it still has major issues to be addressed as follows.

1. The manuscript is poorly written and difficult to read; there are many grammatical errors.
2. The conclusion section contains too many speculations; for example, the discussion about entropic penalty upon contraction in the absence of GCDCA is absurd. I suggest that the authors remove the speculation completely.
3. One major finding in this manuscript is that koff values of murine and human proteins differ significantly. However, koff rates were measured in two different pHs using different methods. I am not convinced that the comparison of koff values is reasonable, although I agree that the koff values may differ to some extent between the two proteins.
4. The authors claim that the CSP of A380 is significant; however, the significance of the CSP was not tested. The escape mutations V378F and L386F are in different positions. Moreover, apo resonances for V378 and L386 were not observed. The authors speculated that the apo resonances are shifted >30 Hz relative to the ligand-bound state. However, there could be other reasons. For example, the apo resonances could be broadened due to conformational exchange; note that this exchange does not need to be between apo and bound states. Instead, they might exchange between apo monomer and apo dimer.
5. The authors argue that binding of GCDCA "causes" loop reorientations in the discussion. What is the evidence that GCDCA causes loop reorientation? It seems that neither causation nor loop reorientation by GCDCA was provided.
6. The authors use "cause" in multiple places. Unless they provide strong evidence, they have to change it to a different word.
7. Fig S16 shows overlaid structures of apo and ligand-bound MNV p-domain. However, it seems that 6lq6.pdb is not the apo MNV P-domain.
8. The authors argue that "CSPs for the methyl groups ... reflect the loop reorientation.." (see Discussion section). However, there is no test in the manuscript showing its evidence. The authors should change it to "may reflect."
9. The crystal structures of the apo MNV P-domain showed the open-close conformations of the EF loop. This indicates that GCDCA simply stabilizes the one conformation. The authors should discuss their findings in light of the pre-existing open-close conformations in the apo protein.

Point-by-point response to reviewer's comments

Reviewer #1 (Remarks to the Author):

I agree with the authors, revised manuscript has improved a lot and addition of native MS data, SEC-MALS data in Fig S5, reprocessing of TROSY spectra, improved structural figures and BMRB assignment provide a lot of clarity. Kudos to the authors for addressing all of the major concerns and additional considerations. Only minor suggestion I will have is the addition of Clustal Omega citation in Fig S2. I advise any of the relevant citation with the following :

Fast, scalable generation of high-quality protein multiple sequence alignments using Clustal Omega Sievers F, Wilm A, Dineen DG, Gibson TJ, Karplus K, Li W, Lopez R, McWilliam H, Remmert M, Söding J, Thompson JD, Higgins D Molecular Systems Biology 7 Article number: 539 doi:10.1038/msb.2011.75

A new bioinformatics analysis tools framework at EMBL-EBI (2010) Goujon M, McWilliam H, Li W, Valentin F, Squizzato S, Paern J, Lopez R Nucleic acids research 2010 Jul, 38 Suppl: W695-9 doi:10.1093/nar/gkq313

Analysis Tool Web Services from the EMBL-EBI. (2013) McWilliam H, Li W, Uludag M, Squizzato S, Park YM, Buso N, Cowley AP, Lopez R Nucleic acids research 2013 Jul;41(Web Server issue):W597-600 doi:10.1093/nar/gkt376

We have included the references to the Clustal Omega algorithm.

Reviewer #3 (Remarks to the Author):

Creutzmacher et al. describe their findings on the monomer-dimer equilibrium of the MNV-P domain and how a ligand affects the equilibrium. Although this revised manuscript is improved compared to the original version, it still has major issues to be addressed as follows.

1. The manuscript is poorly written and difficult to read; there are many grammatical errors.

We have scrutinized the manuscript for grammatical errors and rephrased wordy sentences. We hope the manuscript is now formally correct.

2. The conclusion section contains too many speculations; for example, the discussion about entropic penalty upon contraction in the absence of GCDCA is absurd. I suggest that the authors remove the speculation completely.

We have removed any speculations about entropic penalties.

3. One major finding in this manuscript is that koff values of murine and human proteins differ significantly. However, koff rates were measured in two different pHs using different methods. I am not convinced that the comparison of koff values is reasonable, although I agree that the koff values may differ to some extent between the two proteins.

The dissociation rates differ by about six orders of magnitude, making it impossible to use the same experimental methods for the determination of respective rate constants. We believe that the methods chosen are appropriate and represent the state-of-the-art.

In the meantime, we have extended our studies into dissociation of murine and human norovirus P-dimers. For the murine norovirus P-dimers we have applied dynamic NMR experiments based on methyl TROSY experiments and direct spectra simulation using TITAN. We systematically

studied the influence of pH and bivalent metal ions, and then analyzed the interplay with GCDCA-binding. To obtain more insight into the dissociation of human norovirus P-dimers, we engaged in a collaboration with the lab of Christian Hübner (University of Lübeck, Institute of Physics) and performed single molecule FRET experiments. These experiments yield dissociation rate constants that are in excellent agreement with the data from ion-exchange chromatography reported in the present manuscript.

Briefly, the experiments show that the stability of human and murine norovirus P-dimers increases with decreasing pH. However, the order of magnitude of the respective dissociation rate constants, i.e., s vs. 10^{-6} s, does not change.

To illustrate this point, we show some of our unpublished data. The following figure (Fig. 1) shows a cross-peak of a methyl TROSY spectrum of MNV CW1 apo P-dimers at different pH values (increasing pH from left to right). It is seen that dimer dissociation increases at near-neutral pH values and at very acidic pH values. Fig. 1c illustrates the analysis of dissociation thermodynamics (dissociation constant K_D) and kinetics (dissociation rate constant k_{off}) at a pH of 7.4. The values can be compared to the ones reported in the present manuscript. It is seen that the dissociation rate constant k_{off} is slightly increased at pH 7.4 (ca. 5 s^{-1}) compared to pH 5.3 (this work, ca. 1 s^{-1}). The dissociation constants K_D follow this trend with a value of $45 \text{ }\mu\text{M}$ at pH 7.4 compared to $7 \text{ }\mu\text{M}$ at pH 5.3 (this work).

Fig. 1: pH dependence of P-domain dimer dissociation. (a) Relative intensities in methyl TROSY spectra of monomer and dimer resonances of MNV CW1 apo P-domain are affected by pH, reflecting increased dimer dissociation at very acidic (pD 3) or at near-neutral pD values (pD 5.8 -7.4). (b) This is supported by size exclusion chromatography of P-domain at different pH values. (c) 2D line shape analysis of MNV CW1 apo P-domain methyl TROSY spectra at different protein concentrations and at a pH of 7.4. The line-shape analysis yields a dissociation constant K_D of $45 \pm 1.6 \mu\text{M}$ and a dissociation rate constant k_{off} of $4.7 \pm 0.3 \text{ s}^{-1}$. (Thorben Maas, unpublished data)

The ion-exchange experiments with human norovirus P-dimers (GII.4 Saga) have been performed at pH 7.3 (Fig. 2 this work) yielding a dissociation rate constant k_{off} of $1.5 \times 10^{-6} \text{ s}^{-1}$. We have now performed smFRET experiments with specifically fluorophore labeled GII.4 Saga P-dimers (specific Cys mutants) as seen in the following figure (taken from Carlotta Fiedler, Bachelor Thesis, unpublished).

Abbildung 1.10 Zusammenhang der Transfer-Effizienz mit dem Abstand der Cysteine und Fluorophore. (A) Die Abstände der Cysteine in Homo- und Heterodimeren in der Kristallstruktur der GII.4 Saga P-Dimer N373D Punktmutanten Q366C, V485C und Q504C (PDB: 4X06) wurden in Maestro 12 ermittelt. (B) Je nach Fluorophoren-Paar wird aufgrund der Förster-Radien R_0 verschiedener Fluorophoren-Paare eine unterschiedliche Transfer-Effizienz in Abhängigkeit vom Abstand der Cysteine beziehungsweise Fluorophore erwartet. Das Histogramm (C) zeigt die absoluten Häufigkeiten der apparenten Transfer-Effizienz. Im angegebenen Beispiel Q366C-Q504C ist das Maximum der Gaußverteilung 0,34, was in (B) einem Abstand der Fluorophore (orange) von 58 Å entspricht. Aufgrund der ungefähr 10 Å-langen Verbindung zwischen Cystein und Fluorophore ergibt sich eine Differenz vom Abstand der Cysteine zum Abstand der Fluorophore von circa 20 Å für Dimere, wie für das Beispiel Q366C-Q504C (B) gezeigt wird. (Abbildung (B) modifiziert nach Hirschfeld, V.)

Figure 1.10 Correlation of transfer efficiencies with the distance between cysteines and fluorophores. A) The distances between cysteines in homo- and heterodimers in the crystal structure of the GII.4 Saga P-dimer N373D point mutants Q366C, V485C and Q504C (PDB: 4X06) were determined in Maestro 12. B) Depending on the fluorophore pair, a different transfer efficiency is expected due to the Förster radii R_0 of different fluorophore pairs depending on the distance between the cysteines and fluorophores, respectively. The histogram (C) shows the absolute frequencies of the transfer efficiency. In the given example Q366C-Q504C, the maximum of the Gaussian distribution is 0.34, which in (B) corresponds to a distance of 58 Å between the fluorophores (orange). Due to the approximately 10 Å-long connection between cysteine and fluorophore, the difference between the distance between the cysteines and the distance between

the fluorophores is approximately 20 Å for dimers, as shown for the example Q366C-Q504C (B). (Figure (B) modified after Hirschfeld, V.)

(Carlotta Fiedler, Patrick Ogrissek, and Verena Hirschfeld, unpublished results. Data taken from the bachelor's thesis of Carlotta Fiedler.)

This setup allows systematic studies into the pH-dependence of dissociation constants and dissociation rate constants of Saga P-dimers. We find a dissociation rate constant of 3×10^{-6} Hz at a pH of 7.6, which matches well with the value reported in the present work obtained via IEX. This is ongoing work and experiments at pH 5.3 show that the dissociation rate is slowed down. We have no quantitative data yet from sm FRET at this pH but during these systematic studies we also repeated IEX at pH 5.3, yielding a dissociation rate constant k_{off} of $2.7 \pm 0.2 \times 10^{-7}$ Hz (Patrick Ogrissek, unpublished results).

To summarize, stabilities of human and murine norovirus P-dimers differ by ca. six orders of magnitude independent of pH. All these results are parts of new manuscripts.

4. The authors claim that the CSP of A380 is significant; however, the significance of the CSP was not tested.

For A380 we have measured a CSP ($\Delta V_{\text{Eucl.}}$) of 14 Hz (Table S3). In the following we briefly explain why this CSP is "significant". We have also added three additional figures (new figures Figs. S16 and S17) to the supplementary material illustrating the calculation of a minimum threshold for significant chemical shift perturbations.

Many studies have addressed the issue of precision and accuracy of CSP measurements. One more recent study reports a peak precision of 4 ppb at 900 MHz in gsHSQC $^1\text{H},^{13}\text{C}$ correlation spectra (Arbogast, L.W., Brinson, R.G., and Marino, J.P. (2015). Mapping monoclonal antibody structure by 2D ^{13}C NMR at natural abundance. *Anal Chem* *87*, 3556-3561.), which translates into a peak precision of 3.6 Hz. In our study on human norovirus P-dimers (Mallagaray, A., Creutzmacher, R., Dulfer, J., Mayer, P.H.O., Grimm, L.L., Orduna, J.M., Trabjerg, E., Stehle, T., Rand, K.D., Blaum, B.S., *et al.* (2019). A post-translational modification of human Norovirus capsid protein attenuates glycan binding. *Nat. Comm.* *10*, 1320.) we have experimentally determined the minimum threshold for significant chemical shift perturbations in $^1\text{H},^{15}\text{N}$ TROSY HSQC spectra as 8 Hz (p. 23 ff of the supplementary material of Mallagaray *et al.* 2019). We have repeated this analysis for the present case of $^1\text{H},^{13}\text{C}$ HMQC spectra of the P-domain of MNV CW1. The analysis is now included with the supplementary material (new figure Fig. S16) yielding a minimum threshold of 5.1 Hz for significant CSPs. We have arbitrarily chosen a value of $\Delta V_{\text{Eucl.}} = 10$ Hz for CSPs that are used for further discussion in the main text (see also new figure Fig. S17).

The escape mutations V378F and L386F are in different positions. Moreover, apo resonances for V378 and L386 were not observed. The authors speculated that the apo resonances are shifted >30 Hz relative to the ligand-bound state. However, there could be other reasons. For example, the apo resonances could be broadened due to conformational exchange; note that this exchange does not need to be between apo and bound states. Instead, they might exchange between apo monomer and apo dimer.

We had been aware of this possibility and apologize that we only mentioned it in the supplementary material as part of the legend to Fig. S15. We have now mentioned the possibility of slow exchange processes that are not necessarily linked to exchange between apo and bound states in the main text. We have also edited Table S3 and replaced the statements ">30Hz" with a "#", providing the possible explanations in a footnote.

5. The authors argue that binding of GCDCA “causes” loop reorientations in the discussion. What is the evidence that GCDCA causes loop reorientation? It seems that neither causation nor loop reorientation by GCDCA was provided.

CryoEM studies have shown that the loop orientations are different in the GCDCA-bound and in the apo state. We are reporting CSPs that are in line with these observations. In other words, the data suggest, that CSPs are due to the loop reorientations as observed with cryoEM. Of course, CSPs can have a different cause. Whether it is appropriate to say GCDCA-binding is "causative" for the loop-reorientation or not is difficult to judge. We have rephrased the corresponding text parts.

6. The authors use “cause” in multiple places. Unless they provide strong evidence, they have to change it to a different word.

We have removed the word "cause" almost completely and phrased corresponding passages more carefully.

7. Fig S16 shows overlaid structures of apo and ligand-bound MNV p-domain. However, it seems that 6lq6.pdb is not the apo MNV P-domain.

This is now corrected (3lq6).

8. The authors argue that “CSPs for the methyl groups ... reflect the loop reorientation..” (see Discussion section). However, there is no test in the manuscript showing its evidence. The authors should change it to “may reflect.”

We have changed this (see also point 5 above).

9. The crystal structures of the apo MNV P-domain showed the open-close conformations of the EF loop. This indicates that GCDCA simply stabilizes the one conformation. The authors should discuss their findings in light of the pre-existing open-close conformations in the apo protein.

It is likely that open and closed conformations of the E'F' loop exist in an equilibrium mixture in aqueous solution although this would be difficult to prove. GCDCA-binding would then shift that equilibrium towards the closed conformation. We agree with the reviewer that it is more appropriate to discuss the experimental data considering a pre-existing open-close equilibrium rather than suggesting a static picture where the loop is either open or closed. We have modified the discussion accordingly.